# Variation of CCN activity during new particle formation events in the North China Plain

N. Ma[1,2,4], C. S. Zhao[2,*], J. C. Tao[2], Z. J. Wu[3], S. Kecorius[1], Z. B. Wang[1,4], J. Größ[1], H. J. Liu[2], Y. X. Bian[2], Y. Kuang[2], M. Teich[1], G. Spindler[1], K. Müller[1], D. van Pinxteren[1], H. Herrmann[1], M. Hu[3], A. Wiedensohler[1]

[1]Leibniz Institute for Tropospheric Research, Leipzig 04318, Germany
[2]Department of Atmospheric and Oceanic Sciences, School of Physics, Peking University, Beijing 100871, China
[3]College of Environmental Sciences and Engineering, Peking University, Beijing 100871, China
[4]Multiphase Chemistry Department, Max Planck Institute for Chemistry, Mainz 55128, Germany

*Correspondence to*: C. S. Zhao (zcs@pku.edu.cn)

**Abstract.** The aim of this investigation was to obtain a better understanding of the variability of the cloud condensation nuclei (CCN) activity during new particle formation (NPF) events in an anthropogenically polluted atmosphere of the North China Plain (NCP). We investigated size-resolved activation ratio as well as particle number size distribution, hygroscopicity and volatility during a 4-week intensive field experiment in summertime at a regional atmospheric observatory at Xianghe. Interestingly, based on a case study, two types of NPF events were found, in which the newly formed particles exhibited either a higher or a lower hygroscopicity. Therefore, particle CCN activity differed significantly in those NPF events, indicating that a simple parameterization of particle CCN activity during NPF events over the NCP might lead to poor estimates. For a more accurate estimation of the potential CCN number concentration ($N_{CCN}$) during NPF events, the variation of CCN activity has to be taken into account. Considering that a fixed activation ratio curve or critical diameter are usually used to calculate $N_{CCN}$, the influence of the variation of particle CCN activity on the calculation of $N_{CCN}$ during NPF events was evaluated based on two parameterizations. It was found that $N_{CCN}$ might be underestimated by up to 30% if a single activation ratio curve (representative of the region and season) were to be used in the calculation; and might be underestimated by up to 50% if a fixed critical diameter (representative of the region and season) were used. Therefore, we suggest not using a fixed critical diameter in the prediction of $N_{CCN}$ in NPF. If real-time CCN activity data is not available, using a proper fixed activation ratio curve can be an alternative but compromised choice.

## 1 Introduction

Atmospheric nucleation and the subsequent growth of nucleated clusters, a new particle formation event, NPF, is frequently observed in a variety of environments around the world (Kulmala et al., 2004; Kulmala and Kerminen, 2008 and references therein). This is believed to be an important source of cloud condensation nuclei (CCN). However, the contribution of homogeneous nucleation to the CCN number concentration is still not well defined and has therefore gained considerable attention during the last decade (Kerminen et al., 2012 and references therein).

Because the size of a particle plays the most important role in determining its CCN activity (Dusek et al., 2006) many studies use the number concentration of particles larger than a critical diameter, $D_{P,cri}$ ( i.e., an integral of particle number size distribution, PNSD) as a proxy for the number concentration of potential CCN for a certain super saturation (SS). This method yields an average relative contribution of NPF to CCN spanning from few percent to several tens of percent for different supersaturations, SS, and NPF event type (e.g. Laaksonen et al., 2005; Kuang et al., 2009; Asmi et al., 2011; Peng et al., 2014). Without a direct measurement of the CCN number concentration, such parameterization is reasonable to obtain an approximation of the contribution of NPF to CCN. However, large uncertainties might be incurred in those results. Only a few studies have investigated the variation of the CCN number concentration during an NPF event with direct CCN measurement (e.g. Kuwata et al., 2008; Wiedensohler et al., 2009; Sihto et al., 2011).

The diameter of nucleated clusters is around $1 - 2$ nm (Kulmala et al., 2007). To become effective CCN, newly formed particles need to grow about $10^4$ to $10^5$ times in volume (i.e. to about 30 to 50 nm, depending on cloud types). This means that the chemical composition of those particles is almost entirely determined by the condensing material. Although particle size is the parameter of primary importance, chemical composition does modify the PNSD-based CCN determination by affecting the hygroscopicity or solubility of the potential CCN particles. But this effect is only significant for particles at or slightly greater than $D_{P,cri}$. However, if the condensing vapor causing particle growth has a strong surfactant effect that lowers the water vapor accommodation coefficient or diffusion of condensed water during cloud droplet formation from the particle's surface into its volume, then the chemical composition effect on CCN fraction may extend to diameters much larger than $D_{P,cri}$.

A number of studies have investigated the mechanism of the growth of newly formed particles. Low-volatile organic compounds are suggested to play the major role in particle growth (Kulmala et al., 1998; Kerminen et al., 2000, Wehner et al., 2005), which has been proved in many observations in different region (e.g. Laaksonen et al., 2008; Smith et al., 2010; Ehn et al., 2014), while sulfuric acid was also found to play an important role in the growth process (Birmili et al., 2003; Boy et al., 2005). Yue et al. (2010) found two types of NPF events in Beijing in which either sulfate compounds or organic material (OM) can be the dominant species in ultrafine particles. Therefore, the CCN activity of newly formed particles may differ in different source regions and atmospheric cases, depending on the dominant process in the particles growth. This means, the NPF may influence CCN budget not only by modifying particle number size distribution, but also by changing particle chemistry and consequently CCN activity. However, the variation of particle CCN activity during the NPF event has

been seldom discussed in previous studies. Especially in China, no such investigation has been done for regional NPF events.

Measurements of aerosol particle optical properties (Ma et al., 2011, 2012), hygroscopicity (Liu et al., 2011) and CCN activation (Deng et al., 2011) during the Haze campaign in China (HaChi) showed that the North China Plain (NCP) is one of the most anthropogenically polluted regions in the world. With abundant gas phase precursors in the NCP, NPF occurs frequently (Wu et al., 2007; Wang et al., 2013a) and has a clear and distinct effect on the particle concentration (Guo et al., 2014). The high number concentration of aerosol particles in the NCP may have impacts on cloud microphysical properties and precipitation (Zhao et al., 2006a, b; Deng et al., 2009). To have a better understanding of these impacts, it is essential to have a precise parameterization of particle CCN activity in this polluted region. As a first step, we have therefore undertaken to investigate and understand the contribution and influence of NPF on particle CCN activity, based on a case study in summertime in the NCP.

## 2 Measurements and data processing

### 2.1 Observational Site

During July 9th to August 8th in 2013, microphysical and optical properties of aerosol particles over the size range from 10 nm to 10 μm were measured at Xianghe station (39.75 N, 116.96 E, 36 m a.s.l.), a regional atmospheric observatory in the NCP about 50 km southeast from Beijing and 70 km west from Tianjin (Figure 1). The observatory is located close to a small village and about 5 km west of Xianghe city center. The surroundings are farmland and residential areas. The measurements can be assumed to be representative of the regional background aerosol of the north NCP during daytime (09:00 – 18:00 LT). From time to time, an influence of local anthropogenic emission could be identified during nighttime (18:00 – 09:00 LT). More information about the site can be found in Zhang et al. (2016).

Most of the online instruments were located in a measurement container, in which the temperature was maintained at 22 °C. The inlet system consisted of a PM10 inlet (Rupprecht & Patashnick Co., Inc., Thermo, 16.67 l/min), three in-line Nafion dryers (Leibniz-Institute for Tropospheric Research, Germany; Wiedensohler et al., 2013) and an automatic absorption dryer (Leibniz-Institute for Tropospheric Research, Germany; Tuch et al., 2009). This set up ensured a relative humidity in the aerosol sample flow below 30%. In the measurement container, sampled aerosol was directed to separate instruments through stainless steel and/or conductive tubing using an isokinetic flow splitter.

### 2.2 Particles number size distribution

A mobility particle size spectrometer (TROPOS-type scanning mobility particle size spectrometer, SMPS; Wiedensohler et al., 2012), consisting of a Hauke-type medium differential mobility analyzer (DMA, 28 cm effective length) and a condensation particle counter (CPC Model 3772; TSI, Inc., Shoreview, MN USA), was used to measure PNSDs with mobility diameter from 9 to 800 nm with temporal resolution of 5 min. The measurements were performed in compliance

with recently issued guidelines for atmospheric particle size distribution measurements (Wiedensohler et al., 2012). Data evaluation includes a multiple charge correction (Pfeifer et al., 2014), counting efficiency correction of the condensation particle counters (Wiedensohler et al., 1997), and corrections for the diffusion losses in the system and inlet tubing. The mass concentration of sub-80 nm ($m_{[9-80nm]}$) and sub-800 nm particles ($m_{[9-800nm]}$) was also calculated from the measured

particle number size distribution, with a assumed particle density of 1.6 g cm$^{-3}$.

## 2.3 Size-resolved particle activation ratio

The size-resolved particle activation ratio (AR), defined as the ratio of the number of particles in a defined diameter increment which can be activated instrumentally at a certain SS to the total number of particles in that increment, was measured by a DMA-CCNC system. The system consisted of an electrostatic classifier (Model 3080; TSI, Inc., Shoreview,

MN USA) coupled with a condensation particle counter (CPC Model 3772; TSI, Inc., Shoreview, MN USA) and a continuous-flow CCN counter (Model CCN200, Droplet Measurement Technologies, USA; Roberts and Nenes, 2005; Lance et al., 2006). The system is operated in a size-scanning mode. The monodisperse sample flow from the DMA was split into two flows with flow rate of 1.0 l/min and 0.5 l/min. The CPC and CCN counter measured the total particle and the CCN number concentrations over the particle size range from 9 to 300 nm. More details about the system are given in Deng et al.

(2011).

During the measurement the CCN counter was operated at five SS values sequentially, for 20 min at 0.07% and 10 min each at 0.10%, 0.20%, 0.40% and 0.80%. Two complete size scans were made during each SS setting (4 scans for 0.07%), and only the last scan during each SS was used since the CCN counter needs time for temperature stabilization in the column after the SS is changed. The data from CCN counter during the selected final scan was matched with particle number

concentration measured simultaneously by the CPC. The size-resolved particle activation ratio was then inverted using a modified algorithm based on Hagen and Alofs (1983) (Deng et al., 2011, 2012). The time resolution of a full scan (size-resolved activation ratios at 5 SS) is 1 hour. The sheath and sample flow rates were calibrated before the campaign and checked everyday. The SS of CCN counter were calibrated before the campaign and checked at the end of the campaign with monodisperse ammonium sulfate particles (Rose et al., 2008).

To gain better insight into the temporal variation of CCN activity two parameters were derived, the diameter of 50% activation ratio ($D_{P,50}$) and the slope of the activation ratio curve ($S_{50}$) were used. $S_{50}$ was calculated as $\dfrac{0.2}{\log D_{P,60} - \log D_{P,40}}$,

where 0.2 is the activation ratio difference and $D_{P,40}$ and $D_{P,60}$ are the particle diameters at activation ratios of 40% and 60%, respectively.

## 2.4 Particle hygroscopicity

Particle hygroscopic growth factor ($f_g$), defined as the ratio between the particle diameter at a certain relative humidity (RH) and the particle dry diameter, was measured with a hygroscopic tandem differential mobility analyzer (TROPOS-type

HTDMA; Massling et al., 2007) at 87% RH for dry diameters ($D_{P,dry}$) of 50, 100, 150, 200, 250, 350 nm. The time resolution of the full scan covering the 6 sizes was about 50 min. The probability density function of $f_g$ ($f_g$-PDF) is obtained based on the measured distribution function of $f_g$ with the TDMA inversion algorithm developed by Gysel et al. (2009). Calibration with monodisperse ammonium sulfate particles was automatically conducted every 6 hours.

A hygroscopicity parameter $\kappa$ was calculated according to Petters et al. (2007):

$$\kappa = \left(f_g^3 - 1\right) \cdot \left( \frac{1}{S} \exp\left( \frac{4\sigma_{s/a} M_W}{RT \rho_W D_{P,dry} f_g} \right) - 1 \right) \quad (1)$$

where $S$ is the saturation ratio; $\rho_W$ is the density of water; $M_W$ is the molecular weight of water; $\sigma_{s/a}$ is the surface tension of the solution / air interface which is assumed to be equal to the surface tension of the pure water / air interface; $R$ is the universal gas constant; and $T$ is the temperature. The probability density function of $\kappa$ ($\kappa$-PDF) was then derived and used in

this study. As shown in Zhang et al. (2016), the $\kappa$-PDF usually exhibits a bi-model or tri-model shape. In this study, we simply define the particles with $\kappa > 0.1$ as hygroscopic mode particles, and the rest as nearly-hydrophobic mode particles. More details about this measurement and data processing can be found in Zhang et al. (2016). As a reference, a comparison of $\kappa$ derived with SMPS-CCNC measurement and HTDMA measurement is shown in Fig. S1 in supplement.

## 2.5 Particle volatility

Particle thermal volatility shrinkage factor ($f_s$), defined as the ratio between the diameter of a particle after being heated at a certain temperature and its original diameter, was measured with a volatility tandem differential mobility analyzer (TROPOS-type VTDMA; Philippin et al., 2004) at 300 °C. The same dry particle diameters as HTDMA measurement (50, 100, 150, 200, 250, 300 and 350 nm) was selected for VTDMA measurement. The time resolution of full scans was about 50 minutes.

The VTDMA has a similar structure as the HTDMA with the only difference in the conditioning unit – the humidifier is replaced with a volatilization column, where volatile compounds would evaporate at 300 °C revealing non-volatile particles or cores (Burtscher et al., 2001). The residence time of the particles in the heating column was 0.5 s, which is sufficient to evaporate the volatile fraction of particles in a narrow size range (Philippin et al., 2004). The TDMA inversion algorithm developed by Gysel et al. (2009) was used for data inversion. Ambient temperature (25 °C) scans were used to correct the

size shift between the two DMAs and define the width of the transfer function (Gysel et al., 2009). 203 nm PSL particles were used to calibrate the offset in sizing of the two DMAs on weekly basis.

The probability density function of $f_s$ ($f_s$-PDF) was used in this study. We simply define the particles with $f_s$<0.8 as non-volatile mode particles, and the rest as volatile mode particles. The volume fraction remain of volatile mode particles at 300 °C was calculated with $VFR_V = \dfrac{\int_{f_s=0}^{0.8} f_s \cdot c(f_s) df_s}{\int_{f_s=0}^{0.8} c(f_s) df_s}$ , where $c(f_s)$ is the probability density function of $f_s$.

## 2.6 Particle chemical composition

A Digitel high volume (HV) DHA-80 filter sampler (Riemer, Hausen, Germany) was used to collect atmospheric particles of an aerodynamic diameter less than 10 μm (PM10) on quartz fiber filters (MK 360, Munktell, Falun, Sweden). To reduce the blank content of carbonaceous material, the filters were heated for 24 h at 105 ℃ before sampling. After sampling the filters were stored at −20°C until usage. Samples were taken every 12 h (day time samples: 6:00 – 18:00 LT, night time samples: 18:00 LT until six on the following morning) at a flow rate of 0.5 m$^3$ min$^{-1}$.

Inorganic ions were analyzed by ion chromatography (IC690 Metrohm, Switzerland; ICS3000, Dionex, USA). Before analysis, a filter aliquot was extracted in deionized water by shaking and ultrasonication and filtered through 0.45-mm-pore-size syringe filters.

The determination of total carbon (TC) as organic carbon (OC) and elemental carbon (EC) was carried out by a thermal-optical method using the Sunset Laboratory Dual-Optical Carbonaceous Analyzer (Sunset Laboratory Inc., U.S.A.). For analyses of quartz filters the EUSAAR 2 temperature-protocol was used and a charring correction using light transmission was applied (Cavalli et al. 2010).

## 3 Results and discussion

### 3.1 New particle formation events in the NCP

NPF events have been frequently observed in the north NCP. Based on a 1-year data set, Wu et al. (2007) found NPF events on 40% of measurement days in an urban background station in Beijing. At the regional atmospheric observatory and regional GAW (Global Atmosphere Watch) station Shangdianzi in the north of Beijing, the frequency of NPF day was found to be 36% based on a continuously measurement longer than 1 year (Shen et al., 2011).

During our intensive field campaign, NPF events were observed on 10 of 28 days. Clear smooth growth of nucleation mode particles without strong fluctuations in diameter or concentration (i.e. Class I in the classification system in Dal Maso et al., 2005) was observed on 5 of those days. The time variation of particle number size distribution, $\kappa$-PDF and $f_s$-PDF of 50 nm particles, and the mass fraction of organic matter and sulfate in PM10 during these 5 NPF days are shown in the panels in Fig. 2. During the first NPF event, which occurred on July 20$^{th}$, the nucleation mode did not start at the lower detection limit of our SMPS, meaning that the nucleation event likely occurred upstream of our measurement site. This event was however counted as an NPF event, since the hygroscopicity and CCN activity of the nucleation mode particles during NPF events is

basically determined by the growth process, which was observed on that day. It can be seen that the NPF events started in the morning on all 5 days. In the first 3 cases, the newly formed particles continued to grow through the end of the day. In the other 2 cases, growth was interrupted in the afternoon.

The hygroscopicity and volatility of nanoparticles as measured by the HTDMA and VTDMA is often invoked to provide insight into the particle composition (Zhang et al., 2011). It is interesting to see in Fig. 2 that 50 nm particles exhibited different levels of $\kappa$ in the five NPF events. On July 20[th], 22[nd] and 25[th], after the nucleation mode grew to 50 nm, the $\kappa$-PDF at 50 nm exhibited a dominant hygroscopic mode at $\kappa$ around 0.4 until 18:00 LT. However, the newly formed, 50 nm, particles exhibited a much lower $\kappa$ on July 24[th] and 28[th]. In the late afternoon on July 24[th], the hygroscopic mode was located between 0.1 and 0.2. On July 28, the average $\kappa$ of hygroscopic mode was also lower than 0.2. The hygroscopicity parameter $\kappa$ of a particle is mainly determined by its chemical composition (Petters and Kreidenweis, 2007). Therefore, it can be assumed that the particulate matter produced during the growth process was dominated by different species in different NPF events.

The difference in chemical composition of new particles was also reflected by the VTDMA measurement. Non-volatile residual at 300 ℃ can be found in 50 nm newly formed particles in all the five NPF events. This phenomenon was also observed in Melpitz, Germany (Wehner et al., 2005) and Hyytiälä, Finland (Ehn et al., 2007). And the non-volatile cores were presumed to be polymer-type organics (Wehner et al., 2005; Ehn et al., 2007). In our measurements, the sizes of the non-volatile cores of 50 nm new particles were however different in the five NPF events. In the event on July 20[th], 22[nd] and 25[th], the majority of 50 nm new particles exhibited a shrinkage factor of about 0.3; while on July 24[th] and 28[th], the shrinkage factor is a slightly higher, about 0.4. This means more polymer-type organics were formed during the growth of new particles on July 24[th] and 28[th].

In a study combining in-situ measurements and aerosol dynamic model, sulfuric acid was found to be the major contributor of the growth of newly formed particles in the north NCP, and organic compounds were also found to play a major role in some cases (Yue et al., 2010). Yue et al. (2010) classified NPF events in the north NCP into two types, i.e. sulfur-rich and sulfur-poor events, in which the growth of the new particles is respectively dominated by sulfates and organics. Observed in the same region, our measurements of particle hygroscopicity and volatility can be also well explained by the two NPF types as in Yue et al. (2010). The NPF events on July 20[th], 22[nd] and 25[th] are very like the sulfur-rich type NPF, i.e., condensation and neutralization of sulfuric acid contributed most to the growth of the new particles, and resulted in a high particle hygroscopicity. The NPF events on July 24[th] and 28[th] are like the sulfur-poor event, i.e., condensation of organic compounds had a higher contribution to the growth, resulting in a lower particle hygroscopicity, and more polymers might be produced (Kalberer et al., 2004), resulting in a higher shrinkage factor.

As a reference, the mass fraction of organic and sulfate compounds in PM10 obtained from offline analysis of 12-h DIGITEL HV-samples are shown in the bottom subplots of Fig. 2. It should be noted that the ultrafine particles account for only a minor fraction in PM10 total mass and the mass fractions of organics and sulfate are determined not only by the activity of secondary production, but also the long range transportation and vertical mixing. It can be seen that the average

mass fraction of organics during the daytime of July 20[th], 22[nd] and 25[th] was lower than the average of the previous nighttime, while the mass fraction of sulfate obviously increased (July 22[nd] and 25[th]) or at least stayed at the same level (July 20[th]). And the opposite variation can be found on July 24[th] and 28[th]. This is basically consistent with the measurements of particle hygroscopicity and volatility.

The possible difference in the chemical composition of new particles might be cause by several factors, e.g. the concentration of precursors and the activity of some reactions. Furthermore, these factors might be determined or influenced by other parameters, e.g. ambient temperature, radiation, air mass origin, and vertical mixing. To find out the reason of the composition variation of the new particles is out of the scope of this study, and needs some additional measurements which are not available. In the following sections, we will focus on the CCN activity of new particles in the two types of NPF event.

Since we have no direct measurement of the chemical composition of ultrafine particles, to be accurate, the events on July 20[th], 22[nd] and 25[th] are termed MH-type NPF (NPF with More Hygroscopic particles), while the events on July 24[th] and 28[th] are termed LH-type NPF (NPF with Less Hygroscopic particles).

### 3.2 Variation of CCN activity during the NPF events

Newly formed particles may grow to CCN-active sizes of 50 nm in a few hours and contribute to the total CCN number
concentration (e.g. Wiedensohler et al., 2009; Yue et al., 2011; Wang et al., 2013b; Wu et al., 2015). As discussed above, the hygroscopicity, volatility and chemical composition of aerosol particles might differ in different NPF events in the NCP. The CCN activity of aerosol particles and the CCN productivity of NPF might therefore also differ. As a case study, the NPF events on July 22[nd] (MH-type NPF) and 24[th] (LH-type NPF) are selected and discussed in detail in this section.

### 3.2.1 CCN activity in MH-type NPF

Figure 3 illustrates details of the NPF event that occurred on July 22[nd]. July 22[nd] was a hazy day with average temperature, RH and wind speed of 26.1 ℃, 77.5% and 0.4 ms$^{-1}$, respectively. The weak south-southwest wind continuing from the previous day facilitated the accumulation of pollutants in the region (Xu et al., 2011). The daily average BC mass concentration was 6.94 μgm$^{-3}$, 50% higher than the average 4.66 μgm$^{-3}$ of the entire measurement period.

As the boundary layer developed in the morning, the particulate condensational sink (Kulmala et al., 2001) and the mass
concentration of BC and sub-800 nm particles started to decrease at 07:00 LT. The newly formed particles started to be visible in our SMPS record at around 09:00 LT. The indicatory parameters for NPF, $N_{[40-60nm]}$ and the geometric mean diameter of nucleation mode, increase in coincidence over a three-hour period. A large amount of secondary particulate matter was produced during the growth of new particles, shown as a sharp increase in sub-80 nm particle mass concentration. The new particles continued growing until the end of the day with an average growth rate of 6.3 nm h$^{-1}$ (Fig. 3A). The newly
formed particles reached 50 nm at around 10:30 LT, resulting in a sharp elevation of $N_{[40-60nm]}$ from about $2 \times 10^3$ cm$^{-3}$ to $1 \times 10^4$ cm$^{-3}$. Correspondingly, the number fraction of nearly hydrophobic mode particles ($NF_{NH}$) and non-volatile mode particles ($NF_{NV}$) decreased from about 0.2 to 0, meaning that the newly formed particles grew to 50 nm and dominated the

nuclei mode number. This can also be confirmed by the average $\kappa$-PDF and $f_s$-PDF during the NPF event (Fig. 4). Both $\kappa$-PDF and $f_s$-PDF exhibited narrow unimodal patterns, indicating that the majority of 50 nm particles originated from the same source, NPF. Identifying as MH type, during the NPF event, the 50 nm new particles exhibited a much higher $\kappa_{ave,H}$ than the pre-existing particles (about 0.45 vs. 0.3). As discussed in section 3.1, it is very likely that the condensation and neutralization of sulfuric acid contributed most to the growth of the new particles.

During this NPF event, an enhancement of aerosol CCN activity can be seen. As the $N_{[40-60nm]}$ increased sharply at around 10:30 LT, $D_{P,50}$ decreased from about 46 nm to 39 nm for 0.80% SS, and decreased from about 70 nm to 60 nm for 0.40% SS (Fig. 3F). The $S_{50}$ increased from about 3.5 to 6.0 for 0.80% SS and increased from about 4 to 6 for 0.40% SS (Fig. 3G), meaning that the size-resolved activation ratio curve became steeper. It is interesting that enhancement of aerosol CCN activity can also be seen for 0.20% SS, for which $D_{P,50}$ is larger than the size range dominated by the newly formed particles (Fig. 3C). This is because  low volatility, water soluble compounds produced by gas phase reactions may condense on all particles. The CCN activity of pre-existing particles might therefore also increase.

To have a better view of particle CCN activity during the NPF event of July 22$^{nd}$, three records of size-resolved activation ratio before and during the NPF event are selected and averaged (the corresponding records are marked as color-filled points in Fig. 3F), as shown in Fig. 5. The activation ratio before the nucleation is basically the same as the campaign average at all three SS. However, the activation ratio curves obviously shifted towards lower size and became steeper during the NPF event for SS of 0.40% and 0.80%, indicating that the particles were more hygroscopic, and had a narrower probability distribution of hygroscopicity compared with the pre-existing particles (Su et al., 2010).

In the nighttime after 18:00 LT, due to the collapse of the boundary layer and the increase of aerosol emission (traffic and cooking), the influence of anthropogenic emission starts to be visible in the time series of particle number size distribution (Fig. 3A). The newly formed particles grew further largely  through coagulation and condensation of the freshly emitted particulate and gaseous pollutants, and therefore became less hygroscopic. The BC mass concentration increased significantly after 18:00 LT and peaked at about 20 $\mu gm^{-3}$ at around 22:00 LT (Fig. 3B), resulting in an increase of the number fraction of hydrophobic mode particles (Fig. 3D and S2). These results indicate that in the nighttime a major fraction of the particle population became less hygroscopic, and that different compounds were inhomogeneously distributed among particles. Accordingly, particle CCN activity varies a lot in this period (Fig. 3F and G). Compared with the campaign-average, the activation ratio curve in nighttime is flatter and shifted towards larger size. And the activation ratio reaches only about 80% even at the size of $D_{P,50}\times2$, which is probably due to the high concentration of externally mixed BC particles (also shown as a clear near-hydrophobic and non-volatile mode at 100 and 150 nm in Fig. S2 and S3). Considering the increasing BC mass concentration, the BC emission is very likely to play a major role in the variation of CCN activity in nighttime. Condensation of insoluble organic compounds might be also responsible.

### 3.2.2 CCN activity in LH-type NPF

Figure 6 displays another NPF event that occurred on July 24[th]. This day was relatively clean with a cloudless blue sky. The daily average temperature and RH were 28.4 ℃ and 70.5%, respectively. The BC mass concentration stayed at a low level (1.74 $\mu gm^{-3}$) during the daytime due to the 2 $ms^{-1}$ northwest wind and the development of a deeper boundary layer.

5   The NPF event started at around 09:00 LT. The growth of newly formed particles continued throughout the day with an average growth rate of 6.3 nm $h^{-1}$ (Fig. 6A). It can be seen that $N_{[40-60nm]}$ increased from about $2\times10^3$ $cm^{-3}$ to $1.4\times10^4$ $cm^{-3}$ in a few hours, $N_{[60-80nm]}$ also increased during the daytime. As the newly formed particles grew to 50 nm and became the majority at this size, the number fraction of nearly-hydrophobic mode and non-volatile mode particles decreased to almost 0 within 1 hour. The $\kappa$-PDF and $f_s$-PDF also exhibited narrow unimodal patterns (Fig. 4). It is interesting to note that, unlike the event on July 22[nd], the average $\kappa$ for the hygroscopic mode of 50 nm particles decreased a bit after the nucleation, and stayed below 0.3 for the rest of the day. Such low hygroscopicity of newly formed particles implies that the driving mechanism of particle growth in daytime for this event was somehow different from the event on July 22[nd]. It very likely that sulfuric acid played a less important role in the growth process, and organics had a higher contribution, compared with the MH-type NPF. It can be also seen in Fig. 6E that $VFR_V$ during this NPF event is higher than that during the event on July 22[nd], meaning that more of the polymer-type organics were produced during the growth of the new particles in this event. Correspondingly, no enhancement can be found in particle CCN activity in daytime (Figure 6F and G). $D_{P,50}$ for 0.20% SS even increased from about 110 nm to 120 nm. The average size-resolved activation ratio for 0.20%, 0.40% and 0.80% SS at selected time (marked as color-filled points in Fig. 6F) are shown in Fig. 5. In contrast to the event on July 22[nd], due to the decrease of particle hygroscopicity, the average activation ratio curves shift a bit towards larger diameter during the new particle formation event.

20   The nighttime story on July 24[th] is similar to that on July 22[nd]. The increasing anthropogenic emission caused a decrease in particle hygroscopicity and CCN activity (Figure 6D, F and G). However, the activation ratio at size range of 100 − 200 nm is much lower than that in the nighttime of July 22[nd] (Fig. 5). This is probably because the concentration of background aerosol particles was lower on July 24[th] (also can be seen from the $m_{[9-800nm]}$ in panel B of Fig. 3 and 6). The relative contribution of BC particles in the evening was therefore higher, resulting in a high number fraction of externally mixed BC particles. Nocturnal nucleation also occurred on July 24[th] (Fig. 6A), which is discussed in detail in Kecorius et al. (2015).

### 3.3 Influence of the varying CCN activity on the CCN prediction during NPF events

One of the aims of studying particle CCN activity is to predict the CCN number concentration ($N_{CCN}$) which is defined as the number concentration of particles which can be activated at a certain SS; $N_{CCN}$ can be calculated as follows:

$$N_{CCN} = \int\limits_{\log D_P} AR\left(\log D_P, SS\right) \cdot n\left(\log D_P\right) \cdot d\log D_P$$

30                             (2)

where, $AR$ is the activation ratio function and $n(\log D_P)$ is the particle number size distribution. It can be seen from Eq. (2) that $N_{CCN}$ is determined by both the size-resolved activation ratio and particle number size distribution. Since CCN activity mainly depends on particle size (Dusek et al., 2006), $N_{CCN}$ is more sensitive to the variation of particle number size distribution (Deng et al., 2013). Also, particle CCN activity and hygroscopicity are more difficult to measure and thus to be widely applied. Therefore, a fixed size-resolved activation ratio curve or critical diameter (normally averaged over a certain time period) is usually used when evaluating the contribution of NPF to $N_{CCN}$. However, as discussed in Sect. 3.2, the activation ratio curve can vary largely on NPF days in the NCP. Moreover, the activation ratio curve might be different for different types of NPF events. The questions then arises as to whether we can simply use an average activation ratio curve or a fixed critical diameter for NPF events, and what is the possible bias if an average activation ratio curve or a fixed critical diameter is applied rather than a variable, real-time activation ratio curve.

To answer this question, CCN number concentration was respectively calculated with Eq. (2) based on the campaign average activation ratio (shown as the solid black line in Fig. 5A; the calculated CCN number concentration is termed $N_{CCN,AR\text{-}ave}$), campaign average critical diameter (termed $N_{CCN,D\text{-}ave}$) and real-time activation ratio (termed $N_{CCN,ref}$) at 0.80% SS. The real-time particle number size distribution was used in the calculation. The campaign average critical diameter for a certain SS ($D_{P,cri}$) was chosen as follows: first, $D_{P,cri}$ was calculated for each data record by solving

$$N_{CCN,ref} = \int_{\log D_{P,cri}}^{\log D_{P,max}} n(\log D_P) d \log D_P \tag{3}$$

where, $D_{P,max}$ is the maximum diameter of the measured particle number size distribution; then the geometric average of $D_{P,cri}$ for the whole period was calculated and used. The calculated $D_{P,cri}$ is 55.3, 80.6 and 114.9 nm for 0.80%, 0.40% and 0.20% SS, respectively. Without direct measurement of $N_{CCN}$, the $N_{CCN}$ calculated with real-time activation ratio ($N_{CCN,ref}$) was taken as the reference value. The relative difference between $N_{CCN,AR\text{-}ave}$ and $N_{CCN,ref}$ (termed CCNbiasA), and between $N_{CCN,D\text{-}ave}$ and $N_{CCN,ref}$ (termed CCNbiasB), was then evaluated. CCNbiasA and CCNbiasB were respectively calculated as

$$CCNbiasA = \frac{N_{CCN,AR-ave} - N_{CCN,ref}}{N_{CCN,ref}} \tag{4}$$

$$CCNbiasB = \frac{N_{CCN,D-ave} - N_{CCN,ref}}{N_{CCN,ref}} \tag{5}$$

On July 22[nd] (MH-type NPF), as the newly formed particles grew, $N_{CCN}$ at 0.80% SS doubled during the daytime, from about $1.3 \times 10^4$ to $2.5 \times 10^4$ cm[-3]. The CCN activity of aerosol particles increases during the NPF. As shown in Fig 5, the activation ratio curve was steeper and shifted towards smaller size compared with the campaign average curve. CCNbiasA is therefore respectively about -20%, -15% and -10% for SS of 0.80%, 0.40% and 0.20% in the daytime. On July 24[th] (LH-type NPF), the activation ratio curve during particle growth is very similar to the campaign average (0.80% SS), or even moves a bit to

larger size in the afternoon (0.40% and 0.20% SS, Fig. 5). The CCNbiasA for 0.80% SS during daytime is about -8%, which is much less than the event on July 22$^{nd}$. For 0.40% and 0.20% SS, $N_{CCN,AR-ave}$ is even higher than $N_{CCN,ref}$ in the afternoon. When using the average $D_{P,cri}$, the calculated $N_{CCN,D-ave}$ is lower than $N_{CCN,ref}$ in both of these two NPF events. CCNbiasB are respectively about -40%, -30% and -10% for 0.80%, 0.40% and 0.20% SS in the afternoon of July 22$^{nd}$, and -35%, -40% and 10% for the three SS in the afternoon of July 24$^{th}$.

To give a more general result, CCNbiasA and CCNbiasB were calculated for the entire measurement period. Its overall frequency distribution and the 2-dimensional frequency distribution as a function of time of day are shown in Fig. 7. It can be seen in Fig. 7A that when using the average activation ratio, the majority of CCNbiasA are located between -0.1 and 0.1, meaning that the relative bias of $N_{CCN,AR-ave}$ is lower than 10% in most cases. It can be seen from the contour plot that large negative biases, ranging from -0.1 to -0.3, are all located between 12:00 and 18:00 LT. The large negative biases are caused by the increase of particle hygroscopicity in MH-type NPF event as discussed in section 3.2.1. Therefore, it can be concluded that in the NCP, the bias of $N_{CCN}$ calculated with an average activation ratio may vary between -30%,to 10% during NPF evnets depending on the type of NPF; while the bias is mainly within ±10% in daytime of non-NPF days. It should be noted that this conclusion is based on "using an average activation ratio curve which is representative of the period and area studied". Using an inappropriate activation ratio curve (e.g. an average for another season or another region) may result in higher bias in calculated $N_{CCN}$.

The frequency distribution of the relative difference between $N_{CCN,D-ave}$ calculated with the average critical diameter and $N_{CCN,ref}$ is shown in Fig. 7B. It can be seen that the overall frequency distribution is broader than that of $N_{CCN,AR-ave}$ but the majority of the relative bias is still located between -0.1 and 0.1. However, a long tail can be seen on the left side. The underestimation of $N_{CCN}$ in NPF events is clearer in the contour plot: large body of samples locates at relative difference between -0.1 to -0.5 in the afternoon. This indicates that to use an average $D_{P,cri}$ may result in a larger underestimation of calculated $N_{CCN}$ during NPF events in the NCP. This is because using such a stepwise size-resolved activation ratio overestimates $N_{CCN}$ at $D_P > D_{P,cri}$ and underestimates $N_{CCN}$ at $D_P < D_{P,cri}$. If real-time PNSD and activation ratio curve are similar as the average ones, those overestimation and underestimation may compensate. However, during NPF events, the activation ratio curve may shift towards lower size (in MH-type NPF) and particle number concentration in ultrafine size range increases significantly. The underestimation of $N_{CCN}$ in the left side of $D_{P,cri}$ is therefore much higher than the overestimation in the right side of $D_{P,cri}$, and resulting in a large negative CCNbiasB. Therefore, applying a fixed critical diameter in the calculation of $N_{CCN}$ may result in an underestimation up to 50% in NPF events in the NCP. We should also note that the critical diameter used here is actually "the best estimation" which derived from the samples used also in the test. Assuming an arbitrary critical diameter may result in even larger bias in the estimation of $N_{CCN}$.

It is also worth to note that in both fig. 7A and B, there are a group of samples locates at relative difference between 0.1 and 0.3 in the evening, meaning that the $N_{CCN}$ calculated with the average activation curve or critical diameter may sometimes be overestimated by up to 30% in the evening. As discussed in previous sections, this overestimation is basically caused by the increase of (nearly) hydrophobic matters, mostly BC, in the nocturnal boundary layer. Although BC comes from local

anthropogenic emission, such an increase in BC concentration in the evening is actually a regional phenomenon, which was also observed in other studies in the NCP (e.g. Ma et al., 2011; Cheng et al., 2009). In the NCP, due to the very dense population and the developing industry and agriculture, the villages and cities are densely distributed in the region. The BC emitted from residential area in the evening may diffuse to the area around and causes an increase of the regional 5 background of BC concentration.

## 3.4 Discussion

During NPF events, nucleation creates a large number of nuclei particles. The contribution of the NPF to $N_{CCN}$ is mainly a consequence of growth processes, i.e. coagulation and condensation which enlarge the particles to the size range where they readily act as atmospheric CCN. As discussed above, the chemical composition of particles might change during these 10 processes. In other words, besides enlarging the particles to CCN size range, those processes might also modify the particle CCN activity in varying degrees, depending on the chemical and physical composition of the atmosphere.

Two case studies are shown in sections 3.2.1 and 3.2.2; both include a NPF event defined by a pronounced "banana pattern" in the time series of PNSD. But the CCN activity of the newly formed particles was found to be significantly different. Measurements of particle hygroscopicity and volatility suggest that the growth of the newly formed particles in the daytime 15 was likely to be driven by different species, resulting in different levels of CCN activity at a given size and SS. This means that in the NCP, the CCN activity of newly formed particles during NPF events might be largely different. For example, during the MH-type NPF event on July 22$^{nd}$, the activation ratio curve shows a steeper slope with a lower $D_{P,50}$ than the campaign average in the afternoon; while during the LH-type NPF event on July 24$^{th}$, the activation ratio curve in the afternoon is basically similar to the campaign average. These are only two selected cases. There might be also cases with 20 even more hygroscopic particles than on July 22$^{nd}$, or with even less hygroscopic particles than on July 24$^{th}$, or cases in between. Unfortunately, without detailed chemical analysis of the nuclei mode particles, it is not possible to parameterize the event type from the evolution of PNSD alone which is basically the only way to define a NPF event. It thus might be difficult to find a parameterization of size-resolved CCN activity for NPF events in the NCP which is appropriate for all cases.

25 For accurate estimation of $N_{CCN}$ during a NPF event, we should not only focus on the increase of particle number concentration in certain size ranges. The variation of particle CCN activity should be also taken into account. However, we can only get this information from direct measurement CCN activity, or measurements of hygroscopicity or chemical composition, which are all difficult to be widely applied. Therefore an average activation ratio curve or even a fixed critical diameter have usually been used when evaluating the contribution of NPF to $N_{CCN}$ (e.g. Laaksonen et al., 2005; Kuang et al., 30 2009; Asmi et al., 2011; Peng et al., 2014). From the discussion in section 3.3 we learned that the bias of the estimated $N_{CCN}$ during NPF events can be up to 30% if a fixed activation ratio curve (best estimation, representative of the region and season) is used. Using a fixed critical diameter is likely to result in larger underestimation of $N_{CCN}$. The bias can be up to 50% during NPF events. In daytime without NPF, the bias of estimated $N_{CCN}$ can be limited within 10% if either a proper fixed activation

ratio curve or critical diameter is used. Thus, we suggest not using a fixed critical diameter in the prediction of $N_{CCN}$ in NPF seasons. If real-time CCN activity data is not available, using a proper fixed activation ratio curve can be a compromising choice.

As an important source of aerosol particles, NPF events are very likely to have a major contribution to the number of CCN.
Due to the complexity of the nucleation and growth processes and the inhomogeneous spatial distribution of nucleation and consequent growth, at least in the NCP, this contribution has not been well addressed. Wehner et al. (2010) found that NPF might occur at higher altitudes in the residual layer as well as in the mixed or mixing boundary layer. This means NPF might be even more important in CCN budget than we expect from ground based measurements. More cases of NPF with measurements of particle chemical composition and microphysical properties are therefore needed to better address its role.
It would be also interesting to have vertically resolved CCN measurements.

Our measurements only cover a period of about one month. The average activation ratio is proved to be good enough in the estimate of CCN number concentration in non-NPF periods. It is therefore worth to have a long-term (i.e. longer than one year) measurements of size-resolved activation ratio to provide a precise parameterization of CCN activity in different season or airmass types which can be used to estimate CCN number concentration with lower uncertainty.

## 15   4 Conclusion

To study the variation of particle CCN activity during NPF events in the NCP, size-resolved activation ratio as well as other particle physical and chemical properties were measured during a 1-month field campaign at a regional station in the north NCP.

NPF events were observed on 10 out of 28 days, in which clear and smooth growth of newly formed particles were found on
5 days. In the 5 NPF events, newly formed particles exhibited different hygroscopicity and volatility, suggesting that the particle growth might be dominated by different species during different NPF events. Two NPF events, MH-type and LH-type NPF, were selected for case studies. The size-resolved activation ratio curves during the MH-type NPF event showed a steep shape with lower $D_{P,50}$ than the campaign average; while the activation ratio curves during the LH-type event were basically similar to the campaign average. This means that during NPF in the NCP, the CCN activity of newly formed
particles was different during these two events; the aerosol present during the LH-type event was less active as CCN for a given diameter and supersaturation.

To see the influence of assuming a constant CCN activity in the calculation of $N_{CCN}$ in the NCP, $N_{CCN}$ was calculated with the campaign average activation ratio curve and critical diameter, and then compared with the reference values. The bias of the estimated $N_{CCN}$ during NPF events can be up to 30% if a fixed activation ratio curve is used. Using a fixed critical
diameter is likely to result in larger underestimation of $N_{CCN}$. The bias can be up to 50% during NPF events. In daytime without NPF, the bias of estimated $N_{CCN}$ can be limited within 10% if either a proper fix activation ratio curve or critical diameter is used.

We can learn from this study that for the accurate estimation of $N_{CCN}$ during NPF events, one should not only focus on the increase of particle number concentration in certain size ranges. The variation of CCN activity should be also taken into account. It might be difficult to find a simple parameterization of size-resolved CCN activity for NPF events in the NCP, since it may vary a lot from case to case. Without real-time CCN activity data, a proper fixed activation ratio curve or critical diameter can be used to calculated $N_{CCN}$ for non-NPF daytime. But large uncertainties might appear in predicted the $N_{CCN}$ for NPF event, especially if a fixed critical diameter is used.

**Acknowledgment**

This work is supported by the National Science Foundation of China under Grant No. 41590872, the project Sino German Science Center No. GZ663, and the EU project BACCHUS No. 603445.

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

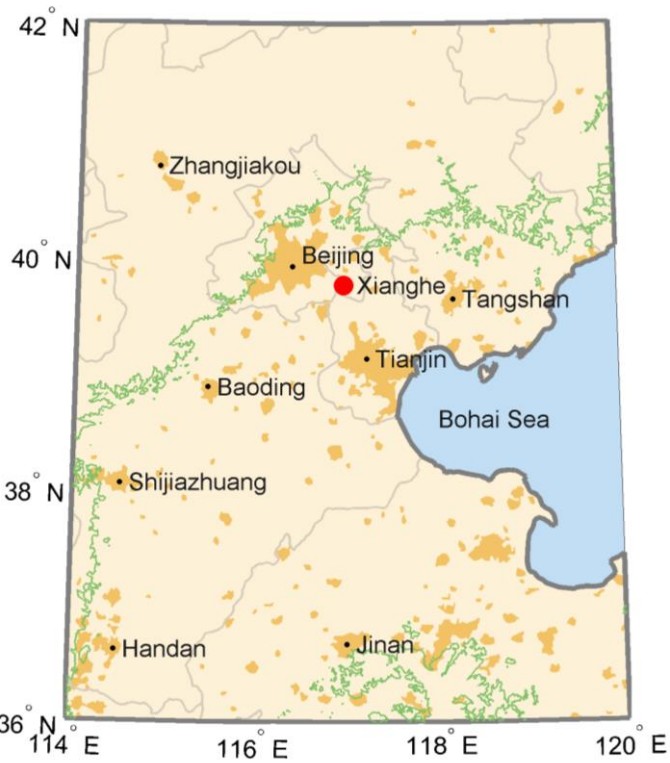

Figure 1. Map of the North China Plain. The observational site is marked as a red point. Urban areas are marked in yellow. The green line denotes the contour line of 500 m a.s.l., which can be considered as the natural boundary of the NCP.

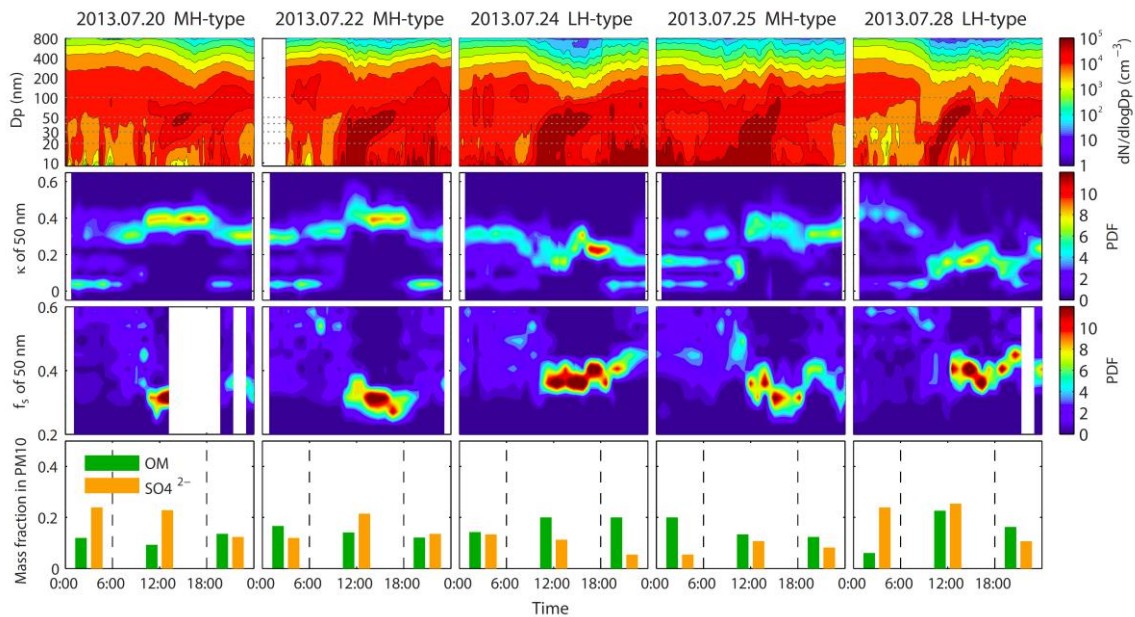

Figure 2. 5 NPF events observed during the campaign period. Subplots show the time series of particles number size distribution, $\kappa$-PDF of 50 and $f_s$-PDF of 50 nm particles, and mass fraction of organics and sulfate from PM10 HV-sample analysis.

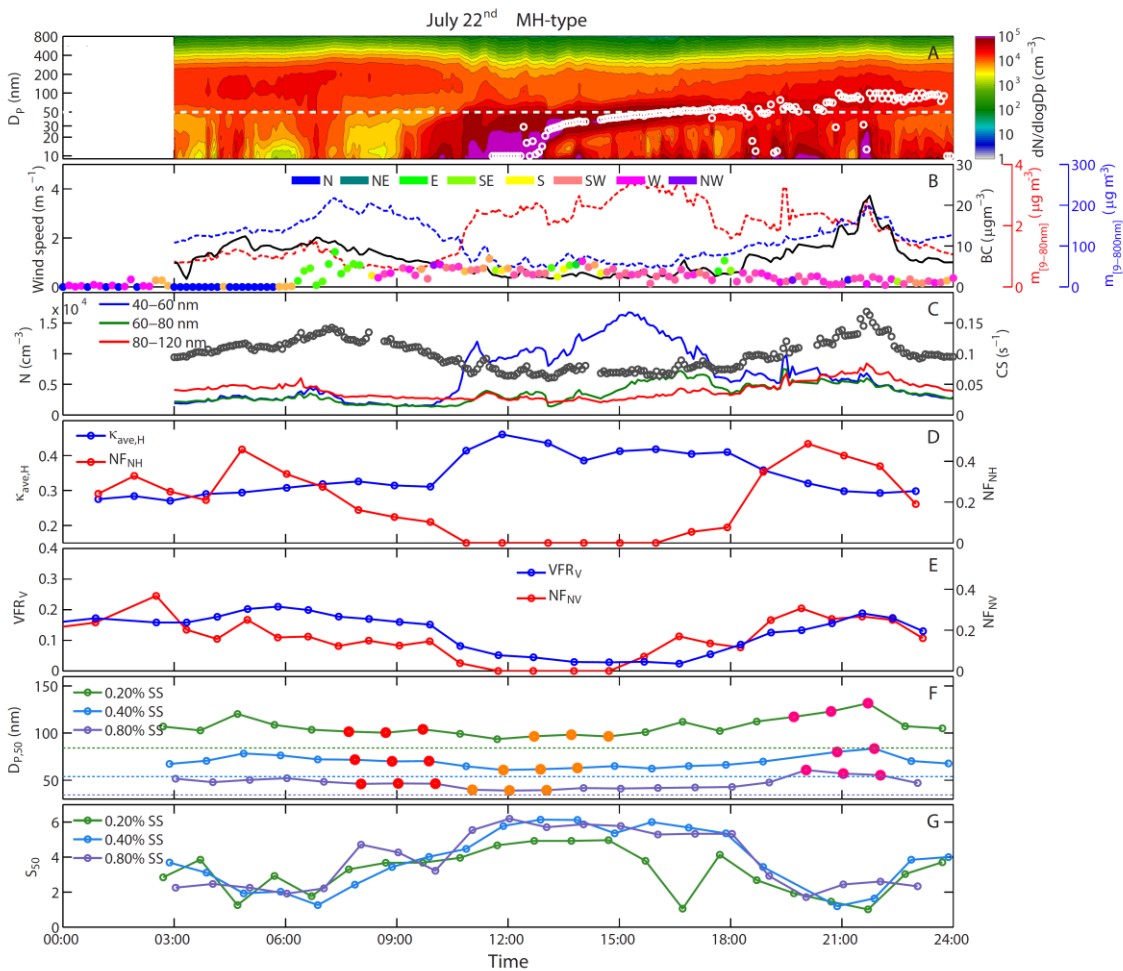

Figure 3. MH-type NPF event on July 22nd. Time series of (A) particle number size distribution and geometric mean diameter of nucleation mode, (B) wind speed/direction and the mass concentration of BC, sub-80 nm and sub-800 nm particles, (C) condensational sink (CS) and number concentration of particles in defined size ranges, (D) average $\kappa$ of hygroscopic mode and number fraction of nearly-hydrophobic mode for 50 nm particles, (E) volume fraction remaining of volatile mode and number fraction of non-volatile mode for 50 nm particles, (F) $D_{P,50}$ for 0.20%, 0.40% and 0.80% SS, as well as (G) $S_{50}$ for the three SS. The dashed lines in panel F show the theoretical critical diameters for ammonium sulfate at the three SS. Points filled with color in panel G show the records selected to calculate the average size-resolved activation ratio shown in Fig. 5.

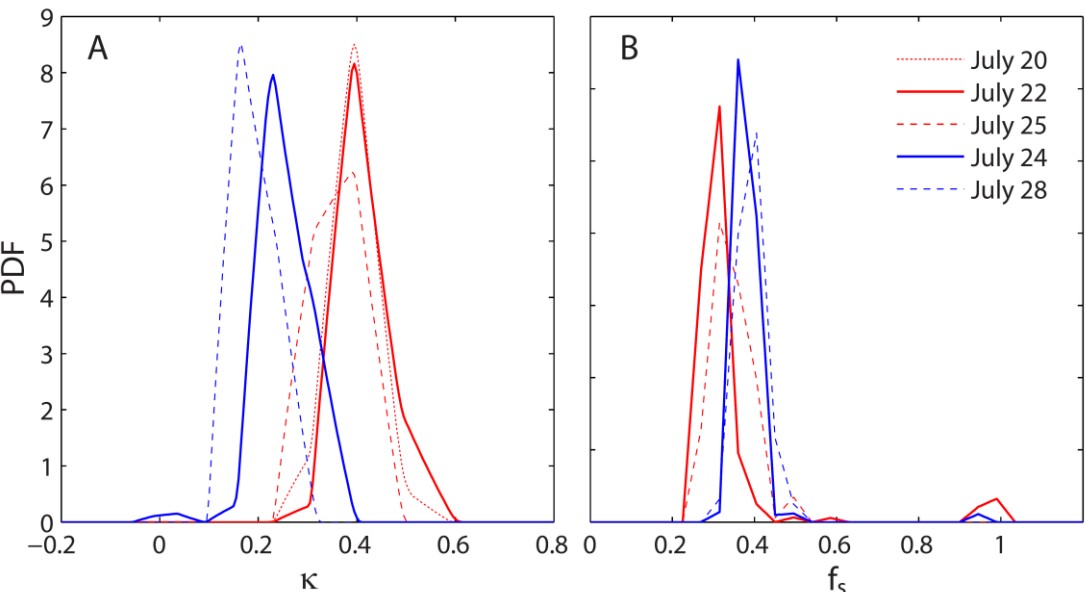

Figure 4. Average $\kappa$-PDF (A) and $f_s$-PDF (B) of 50 nm particles during the 5 NPF events.

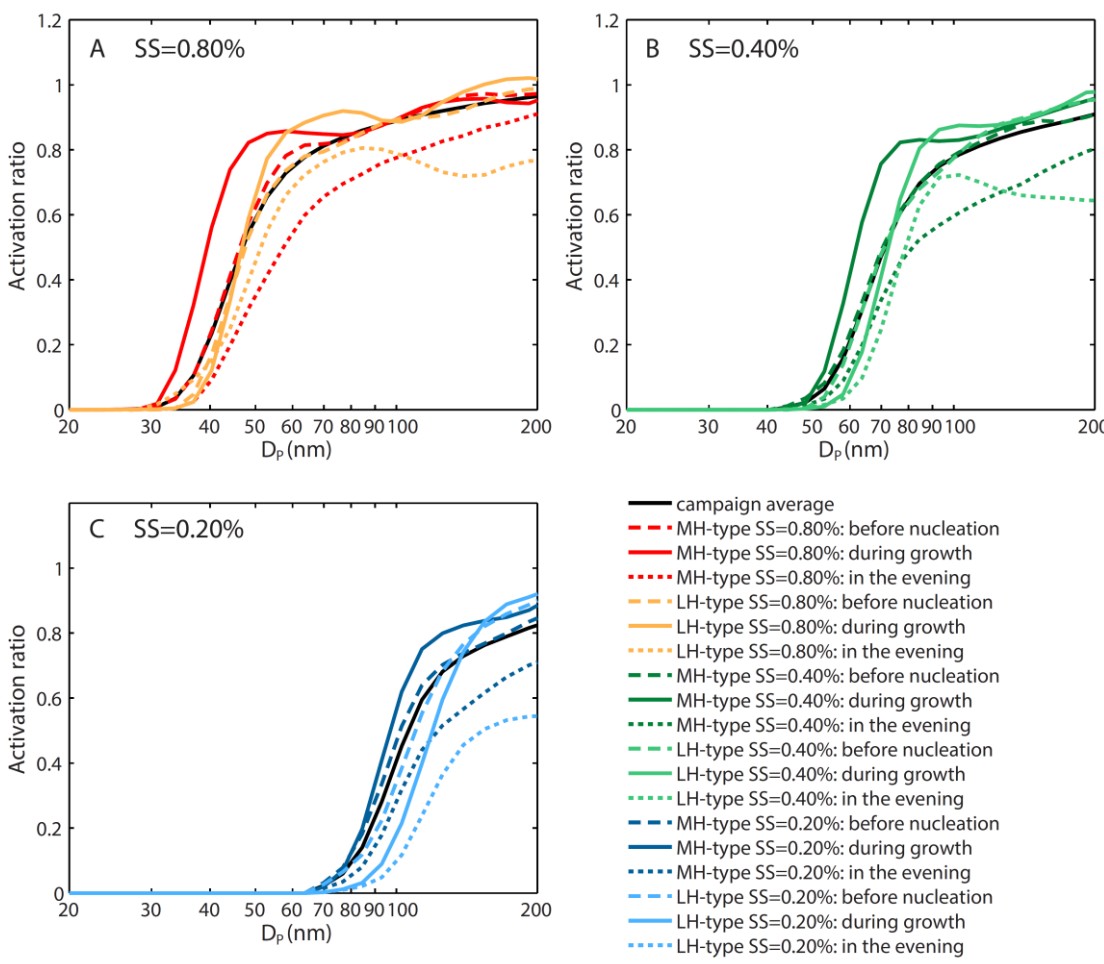

Figure 5. Average size-resolved activation ratio for the selected time periods on July 22$^{nd}$ and 24$^{th}$

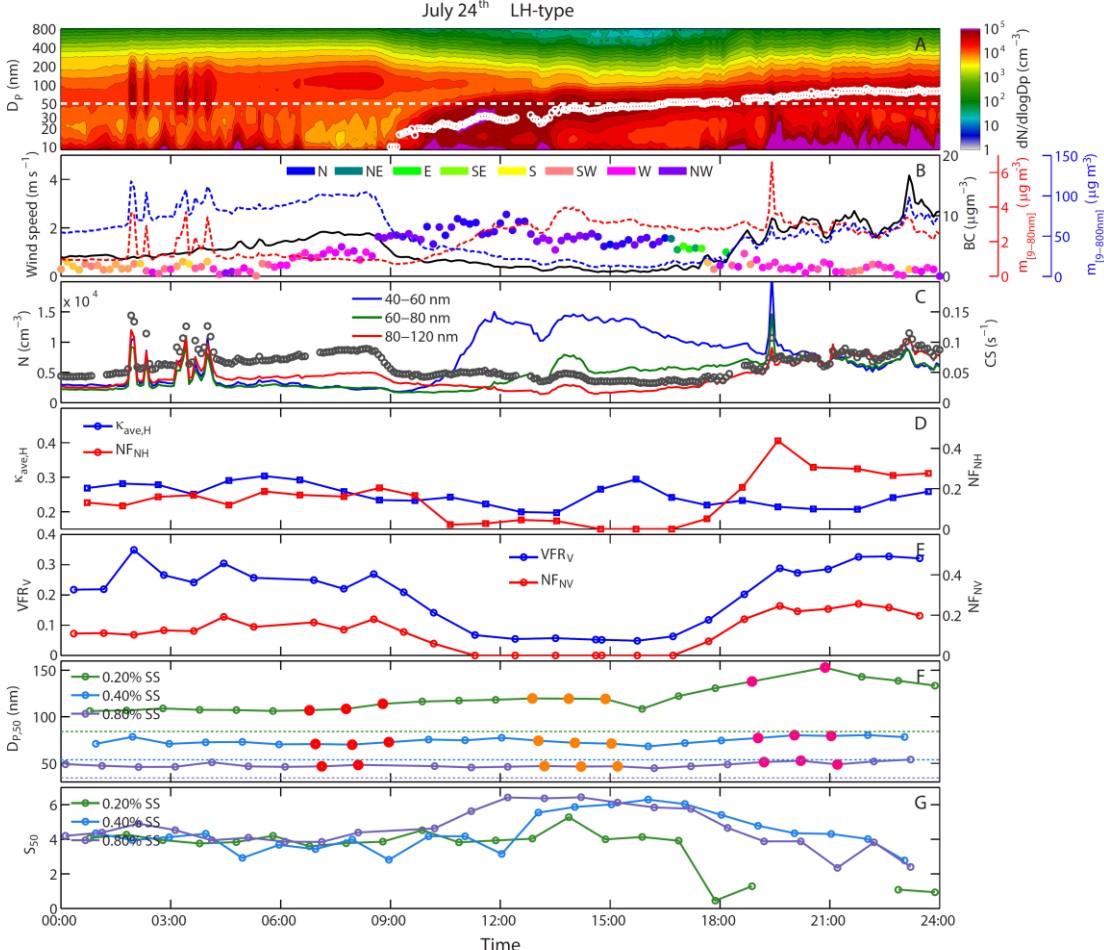

Figure 6. LH-type NPF event on July 24$^{th}$. Time series of (A) particle number size distribution and geometric mean diameter of nucleation mode, (B) wind speed/direction and the mass concentration of BC, sub-80 nm and sub-800 nm particles, (C) condensational sink (CS) and number concentration of particles in defined size ranges, (D) average $\kappa$ of hygroscopic mode and number fraction of nearly-hydrophobic mode for 50 nm particles, (E) volume fraction remaining of volatile mode and number fraction of non-volatile mode for 50 nm particles, (F) $D_{P,50}$ for 0.20%, 0.40% and 0.80% SS, as well as (G) $S_{50}$ for the three SS. The dashed lines in panel F show the theoretical critical diameters for ammonium sulfate at the three SS. Points filled with color in panel G show the records selected to calculate the average size-resolved activation ratio shown in Fig. 5.

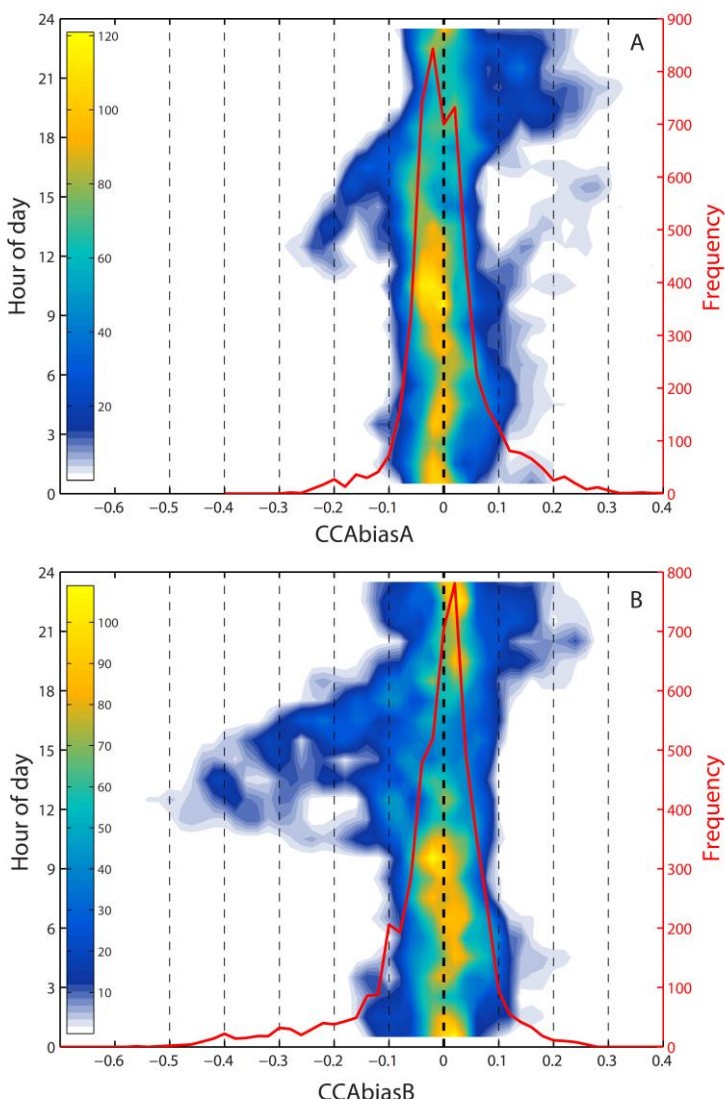

Figure 7. 2-dimensional frequency distribution (number of occurrences per interval of CCNbias and time of day, shown as contour plot) and overall frequency distribution (number of occurrences per interval of CCNbias, shown as red line) of CCNbiasA (panel A) and CCNbiasB (panel B) for the entire campaign period

