# Peer review of "Variation of CCN activity during new particle formation events in the North China Plain"

_Atmospheric Chemistry and Physics, 2016_

## Referee Comment (RC1) · Anonymous Referee #1 · 18 Feb 2016

General comments.

This paper presents a detailed analysis of instrumental measurements of particle physics and their cloud condensation parameters during an intensive field campaign in a highly polluted region of the North China Plain. The analysis focused on the CCN activity potential of the particles in the 20 to 200nm size range. CCN activation ratio data are unique for this region and provide results of value to the wider atmospheric sciences and cloud physics community. Chemical analysis for the size range of particles most important for CCN activity of the total particle population is lacking. The measurement of hygroscopic growth factor helps in understanding the CCN of the several events and indicated a chemical effect but the compounds or groups of compounds causing this effect cannot be identified with certainty. The data presentation via graphs is generally clear, quantitative and provides an excellent integral overview.

Specific comments.

Abstract and introduction: Mention of region and season is made. The data base of the study is different event types in one region, NCP, over one four week time period. Thus, it would be more accurate and clear to refer to the results in terms of a case study of NPF type, i.e., based on your NCP PNSD and chemical composition data for the summertime campaign.

Page 2 line 18: i.e. at least up to 50 nm. This value of critical diameter depends on supersaturation in the parcel as it cools below the dew point temperature by lifting, radiation or mixing. For fog and low level stratus with limited vertical motion the Dcrit may be around 50 nm. For stratocumulus it may be closer to 30 nm. For cumulus, less than 30 nm. Your point about growth by condensation (and some coagulation) determining the relative organic vs. inorganic (with greater vs. lesser water solubility) chemical composition of the CCN is still valid. A volume growth factor of 10ˆ4 to 10ˆ5 is still needed.

Line 20: It should be mentioned that, although particle size is the parameter of primary importance, chemical composition does modify the PNSD-based CCN determination by affecting the hygroscopicity or solubility of the potential CCN particle. But this effect is only significant for particles at or slightly greater than Dcrit . However, if the condensing vapor causing NP growth has a strong surfactant effect that lowers the water vapor accommodation coefficient or diffusion of condensed water during cloud droplet formation from the particle's surface into its volume, then the chemical composition effect on CCN fraction may extend to Dp much larger than Dcrit .

Page 3, line 9: State the particle size range. ... particle size range from xx to xxx nm.

line 12: Was the data during locally influenced time periods eliminated from the analysis? What hours constituted daytime? Does it match the chemical sampling schedule? Was this simply determined by hour of the day or was there meteorological input such as inversion height, thermal stability, wind speed?

Line 15: Does "Inc., Thermo," refer to the nafion dryer?

Page 4, line 4: What was the particle diameter range of the DMA-CCNC system? Does it match the SMPS system? What was the time resolution of the AR measurement system? Was it operated in a scanning or stepping mode? If the latter, what was the step interval?

Line 11 "The size-resolved particle activation ratio was then inverted ... " I understand the general idea here but not the details. You have AR for specific SS and Dp values. Eventually you determine the activation ratio curve, as the function AR(Dp,SS) used in Eqn. 2. The inversion operation is not clear; maybe it is not needed if explained in Deng 2011.

Line 14: Was the SS calibration done with monodisperse particles of known chemical composition?

Line 23: Were the TDMA calibration ammonium sulfate particles monodisperse?

Page 5, Figure 2: The details of the PNSD data that are discussed are not clear to this reader even at 300% magnification due to the high time and size variability of the data. This could perhaps improved by smoothing of the contours. The fine structure in the data is not needed to convey the general features such as "nucleation mode did not start at the lower detection limit of our SMPS", and the beginning and end of particle growth by condensation of vapors. A quantitative label for color scale is needed for panels 1 and 2. The hygroscopic growth at 50 nm is most germane to understanding CCN activation; however $\kappa$-PDF at other diameters would be of interest, perhaps as a supplement to the manuscript.

Page 6, line 10: The inference of the chemistry of nuclei mode particles from PM10 chemical analysis leaves a lot to be desired. Is there data from previous experiments in the NCP when mass spectrometric analysis was done to aid in the understanding of what the size dependence of sulfate and organic compounds might have been in this

field study?

Line 17: I don't understand "production activity". Do you mean rate of condensation?

Line 24: It would be useful in this figure to identify the two events, the sulfate-dominant NPF event and OM-dominant NPF events, either in the caption or under the dates at the top of panel 1.

Line 25: "And two types of NPF ... " Does this refer to Yue's work? If yes, then I suggest, "Furthermore, they observed two types of NPF .... "

Line 31: You present BC mass concentrations. Do you have similar mass concentration data for sulfate and OM? Total sub 800nm and sub 80 nm mass derived from the SMPS volume and an assumed density would be useful as well.

Page 7, line 4, Figure 3A The label on the ordinate should be Dp, particle diameter. The color scale label should be placed next to the color scale bar.

Figure 3 overall: The timing of the appearance of NFP varies depending on the parameter. NFP appear sporadically beginning at 900, Fig 3A. $\Delta N(40,60)$ and $\kappa$ increases sharply at about 1030. Nuclei mode reaches 50nm about 1200LT. The condensational sink does not decrease sharply over time in the morning. Of course there are other parameters not quantified and presented such as advection or mixing from more polluted layers aloft or actinic flux and photochemical precursor gas formation. These are beyond the scope of the observations and discussion. Simply mention that the indicatory parameters for NFP, Dp,nuclei, $\Delta N(40,60)$ and $\kappa$, increase more or less in coincidence over a three hour period.

Figure 3 D: The legend shows NFH while the right hand label shows NFNH.

Line 27, Figure 3E: The highlighting of the points used for calculating the average size-resolved activation ratio shown in Figure 5 is difficult to see and the change in color is not clear. A larger circle for those points would help the reader at a glance.

Also, a theoretical critical Dp line for ammonium sulfate as a reference for the three SS values would be useful for comparison.

Page 8, line 30, Figure 5: What is the SS for the campaign average curve? It is difficult to follow all the curves and colors on this plot. I suggest breaking the single plot into three panels, one for each SS. The information in Figure 5 is summarized in Figure 3 panels E and F, but, these do not show the OM vs. sulfate differences or that activation ratios remain below 1.0 even at sizes much larger than Dcrit,SS, so figure 5 is of value for showing this result.

This last point, AR < 1 at Dp»Dp,crit, needs explanation. Certainly size plays the dominant role as you point out since approximately 80% of the particles activate at Dp50 plus about 5 nm. However, on average only about 85% activation ratio is reached at twice the Dp50. In two cases the maximum AR is only about 60%. Do you have an explanation, e.g., truly insoluble, non-wettable primary particles, or organic surfactants?

Page 8, line 11, Figure 6: Change caption and legends and plotting in figure 6 as suggested for figure 3. See Word changes in figure 3 caption.

Page 9, line 29: Do you know from chemical measurement that sulfate dominated the particle growth size mode, i.e., nuclei and Aitken modes? From PM10 sulfate analysis alone that cannot be supported. The best evidence you have is for sulfate vs. OM is from your $\kappa$ and NFNH.

Page 10, lines 1 through 3: It is not clear how the difference ratio plotted in Figure 7 was calculated, what NCCN ref and NCCN Aravg are. The text and figure caption and x-scale label symbols are not consistent. Provide an equation (4) in the text for the bias parameter used in the figure e.g. CCNbias = ..........

Line 13, Figure 7: The caption should the time period of average, i.e., afternoons of July 22nd and July 24th. I assume the frequency statistics are number of occurrences per interval of relative ratio, and for the 2 dim, number of occurrences per interval of

relative ratio and time. Mention this in the caption.

Are there significant differences between the two parameters? To me they look very similar other than the left hand tail. You do not mention the chemistry or physics behind this tail. Nor can I imagine any.

In other plots and discussions you have presented and compared data from 22 July vs. 24 July. Were there significant differences in the frequency distribution plots for those two days?

Page 11, line 31: " ... bias of the estimated NCCN ranges from about 0 to 30% ... " From your frequency distribution plots in Figure 7, I would put the biases at $\pm 10\%$. That is my simple ocular analysis. Better would be the percent bias at $\pm$ one or two standard deviations of the of the data.

Page 12, line 25: I would make a stronger statement. This means that during NPF in the NCP, the CCN activity of newly formed particles was different during these two events; the aerosol present during the organic dominated event was less hygroscopic and less active as CCN for a given diameter and supersaturation.

Line 29: Again your 0 to 30% values could be more accurately presented as I suggested above, page 11, line 31.

---

## Referee Comment (RC2) · Anonymous Referee #2 · 29 Feb 2016

Reference No.: acp-2016-23 Authors: N. Ma et al. Title: Variation of CCN activity during new particle formation events in the North China Plain

This paper investigated the ability of aerosols acting as CCN during the new particle formation (NPF) events over North China Plain. The authors focused on two different NPF events and suggested that the possible deviation of simplified NCCN estimation might be significant for the NPF periods.

Overall comment: The authors differentiated two cased based on the sulfate fraction in PM10 to conclude the sulfate dominant and OM dominated cases. However, the sulfate fraction in PM10 cannot be representative for sub-micron sized particles and that was also stated by the authors. If sulfate fraction in PM10 can represents for the sulfate fraction in sub-micrometer sized particles, the authors should also expect the

high sulfate fraction should correspond to higher kappa for all periods and this is not always true. For example, the kappa of 7/20 in the early morning showed a much lower value even though the sulfate fraction is similar to the daytime of 7/20. This analytical method might be suspicious without further validation. The composition variation might be due to the parcel source since the provided two cases has the source difference; i.e. west-southwest wind for 7/22 and northwest wind for 7/24. Could the radiation also play a role affecting the hygroscopicity because 7/22 is a hazy day while 7/24 is a sunny day? Overall, it is difficult to conclude the exact factor with only two cases.

Other comments: 1. Lines 4-11 of Page 4: How many sizes did the authors scan for a given SS?

2. There were two methods applied to determine the hygroscopicity. How is the comparison?

3. Line 20 of Page 20: the authors stated there are 10 out of 28 days associated with NPF events but only 5 days with particle growing. What parameter controls the growing process? Does the hygroscopicity of aerosols at these two cases different?

4. Because the authors focused on the NPF events, the date of Figure 4 should be the same as Figure 2.

5. Line 10 of Page 9: What is the average activation ratio profile the authors applied?

6. The label of y axis for all SMPS figures should be Diameter (nm) not the number concentration.

7. The color bar for SMPS and k in Figure 2 should be provided.

8. There are two ticks for the right y axes of Figure 6B, 6C and 6D which should be modified.

---

## Author Comment (AC1) · 23 May 2016

**Response to comments of referee #1**

**General comments**

*This paper presents a detailed analysis of instrumental measurements of particle physics and their cloud condensation parameters during an intensive field campaign in a highly polluted region of the North China Plain. The analysis focused on the CCN activity potential of the particles in the 20 to 200nm size range. CCN activation ratio data are unique for this region and provide results of value to the wider atmospheric sciences and cloud physics community. Chemical analysis for the size range of particles most important for CCN activity of the total particle population is lacking. The measurement of hygroscopic growth factor helps in understanding the CCN of the several events and indicated a chemical effect but the compounds or groups of compounds causing this effect cannot be identified with certainty. The data presentation via graphs is generally clear, quantitative and provides an excellent integral overview.*

**Reply:**

Thanks for the comment.

It was a pity that we have no online measurement of chemical composition of nucleation mode particles during the NPF events. Actually, our collaborator did AMS measurement during the joint intensive campaign. Unfortunately, we met some problem during transporting of our mobile laboratory. Our measurement therefore started too late and has no overlap with AMS measurement in the campaign.

However, we did particle volatility measurement with a volatility tandem differential mobility analyzer (TROPOS-type VTDMA; Philippin et al., 2004) during the campaign. The hygroscopicity and volatility of nanoparticles as measured by the HTDMA and VTDMA is often invoked to provide insight into the particle composition (Zhang et al., 2011). We therefore decided to involve this dataset in our study, together with hygroscopicity data, to provide some additional information about the new particles.

Our study aimed at evaluating the variability of the CCN activity during NPF events in an anthropogenically polluted atmosphere and the applicability of some simplified CCN parameterization. The online measurements of hygroscopicity and volatility of new particles can already provide indication for distinguishing different NPF types. Without direct measurement of chemical composition, to be more accurate, in the revised manuscript, the two NPF types were termed MH-type NPF (more-hygroscopic type NPF) and LH-type NPF (less-hygroscopic NPF), instead of sulfate-dominant NPF and OM-dominant NPF. And we built most of the discussion in section 3.1 and 3.2 on the basis of HTDMA and VTDMA measurements, instead of on PM10 chemical composition data. In the response to the specific comments, there is more detailed information about this.

**Specific comments**

**Reviewer:**

*Abstract and introduction: Mention of region and season is made. The data base of the study is different event types in one region, NCP, over one four week time period. Thus, it would be more accurate and*

*clear to refer to the results in terms of a case study of NPF type, i.e., based on your NCP PNSD and chemical composition data for the summertime campaign.*

**Reply:**

Thanks for the comment.

We have revised the abstract as "…We investigated size-resolved activation ratio as well as particle number size distribution, hygroscopicity and volatility during a 4-week intensive field experiment in summertime at a regional atmospheric observatory at Xianghe. Interestingly, based on a case study, two types of NPF events were found, in which the newly formed particles exhibited either a higher or a lower hygroscopicity…". And in the introduction, we have revised the last sentence as "…we have therefore undertaken to investigate and understand the contribution and influence of NPF on particle CCN activity, based on a case study in summertime in the NCP"

**Reviewer:**

*Page 2 line 18: i.e. at least up to 50 nm. This value of critical diameter depends on supersaturation in the parcel as it cools below the dew point temperature by lifting, radiation or mixing. For fog and low level stratus with limited vertical motion the Dcrit may be around 50 nm. For stratocumulus it may be closer to 30 nm. For cumulus, less than 30 nm. Your point about growth by condensation (and some coagulation) determining the relative organic vs. inorganic (with greater vs. lesser water solubility) chemical composition of the CCN is still valid. A volume growth factor of 10ˆ4 to 10ˆ5 is still needed.*

**Reply:**

Thanks for the comment. The sentence has been revised as: "To become effective CCN, newly formed particles need to grow about $10^4$ to $10^5$ times in volume (i.e. to about 30 to 50 nm, depending on cloud types)"

**Reviewer:**

*Line 20: It should be mentioned that, although particle size is the parameter of primary importance, chemical composition does modify the PNSD-based CCN determination by affecting the hygroscopicity or solubility of the potential CCN particle. But this effect is only significant for particles at or slightly greater than Dcrit . However, if the condensing vapor causing NP growth has a strong surfactant effect that lowers the water vapor accommodation coefficient or diffusion of condensed water during cloud droplet formation from the particle's surface into its volume, then the chemical composition effect on CCN fraction may extend to Dp much larger than Dcrit .*

**Reply:**

Many thanks for the comprehensive explanation. The sentence has been expanded as: "Although particle size is the parameter of primary importance, chemical composition does modify the PNSD-based CCN determination by affecting the hygroscopicity or solubility of the potential CCN particles. But this effect is

only significant for particles at or slightly greater than $D_{P,cri}$. However, if the condensing vapor causing particle growth has a strong surfactant effect that lowers the water vapor accommodation coefficient or diffusion of condensed water during cloud droplet formation from the particle's surface into its volume, then the chemical composition effect on CCN fraction may extend to diameter much larger than $D_{P,cri}$."

**Reviewer:**

*Page 3, line 9: State the particle size range. ... particle size range from xx to xxx nm.*

**Reply:**

The sentence has been revised as "…microphysical and optical properties of aerosol particles over the size range from 10 nm to 10 μm were measured at Xianghe station…"

**Reviewer:**

*line 12: Was the data during locally influenced time periods eliminated from the analysis? What hours constituted daytime? Does it match the chemical sampling schedule? Was this simply determined by hour of the day or was there meteorological input such as inversion height, thermal stability, wind speed?*

**Reply:**

The daytime period is roughly defined as from 09:00 to 18:00, i.e. between the morning rush hour and evening rush hour. This definition is actually based on aerosol measurements. Figure R1 shows the campaign-average diurnal variation of particle number size distribution, BC mass concentration ($m_{BC}$) and single scattering albedo ($\omega$). We can see during the time period between 09:00 and 18:00 LT, the average PNSD is basically dominated by the new particle formation without indication of local emission. BC mass concentration stays in a relatively low level and single scattering albedo is around 0.9. Since the anthropogenic emission is low and the boundary layer is well mixed in daytime in summer, the daytime measurements are assumed to be more representative of the background of the region.

In the 5 days used in this study, we did not find visible influence of local anthropogenic emission on our measurements during daytime. But the influence is always obvious in the evening, as shown in Fig.3 and 6 in manuscript. Since we mainly focused on the new particle formation in daytime, the locally influenced time periods (in nighttime) was not eliminated in the figures.

The chemical sampling always started at 06:00 and 18:00, and lasted for 12 hours. Considering the PM10 chemical composition data is not appropriate for this analysis, we decided to remove most of the content about PM10 data.

The hours constituting daytime and nighttime are now given in the text.

[Figure]

Figure R1. Average diurnal variation of particle number size distribution (upper), single scattering albedo and mass concentration of black carbon (lower).

**Reviewer:**

*Line 15: Does "Inc., Thermo," refer to the nafion dryer?*

**Reply:**

"Rupprecht & Patashnick Co., Inc., Thermo" refers to the PM10 impactor inlet. The Nafion dryer is made by Leibniz-Institute for Tropospheric Research. This information has been added in the text: "…three in-line Nafion dryers (Leibniz-Institute for Tropospheric Research, Germany; Wiedensohler et al., 2013) and…"

**Reviewer:**

*Page 4, line 4: What was the particle diameter range of the DMA-CCNC system? Does it match the SMPS system? What was the time resolution of the AR measurement system? Was it operated in a scanning or stepping mode? If the latter, what was the step interval?*

**Reply:**

The diameter range of the DMA-CCNC system is about from 9 to 300 nm. The diameter range of SMPS (TROPOS-type, for the measurement of PNSD) is from 9 to 800 nm. The reason we only measured up to 300 nm for DMA-CCNC is listed as following.

1) The aerosol flow in DMA is 1.5 lpm. To keep a proper aerosol to sheath flow ratio, the sheath flow is high. Therefore the upper limit of the size range is about 300 nm.

2) We mainly focused on the ascending part of the activation curve. Most particles larger than 300 nm were activated at the applied supersaturation.

The DMA-CCNC system is operated in a size-scanning mode. The time resolution of a full scan (5 supersaturaitons, 2 full-size scan for each supersaturation) is about 1 hour. In the text we have added: "The system is operated in a size-scanning mode" and "The time resolution of a full scan (size-resolved activation ratios at 5 SS) is 1 hour"

**Reviewer:**

*Line 11 "The size-resolved particle activation ratio was then inverted ... " I understand the general idea here but not the details. You have AR for specific SS and Dp values. Eventually you determine the activation ratio curve, as the function AR(Dp,SS) used in Eqn. 2. The inversion operation is not clear; maybe it is not needed if explained in Deng 2011.*

**Reply:**

The inversion algorithm applied in this study is similar like that given in Deng et al. (2011), but some improvement was made. The algorithm in Deng et al. (2011) only corrects the effect of multiple charges. The algorithm applied in this study also considered the width of the DMA transfer function. The improved algorithm has been published in Acta Scientiarum Naturalium Universitatis Pekinensis (in Chinese), this paper (Deng et al., 2012) has been added in the reference. In the following we provide a brief description of the new algorithm.

Assuming the voltage of a DMA is set to $V_i$ $(i = 1, 2, ..., I)$ to select particles with electrical mobility of $Z_{pi}$ $(i = 1, 2, ..., I)$. The CCN number concentrations measured by CCNC are

$$R_i = \int_0^\infty G(i, x) A(x) n(x) dx \quad (1)$$

where, $x = \log D_P$, $A(x)$ is activation ratio, $n(x)$ is the inverted PNSD. Kernel function $G(i, x)$ can be expressed as

$$G(i, x) = \sum_{\upsilon=1}^\infty \phi(x, \upsilon) \Omega(x, \upsilon, i) \quad (2)$$

where, $\phi(x, \upsilon)$ is the probability of a $x$-size particle taking $\upsilon$ elementary charges, and $\Omega(x, \upsilon, i)$ is DMA transfer function (i.e. probability of $x$-size particle with $\upsilon$ elementary charges "surviving" in the output aerosol flow of DMA with voltage of $V_i$).

Now we need to solve equation (1) to get $A(x)$ from measured $R_i$ and $n(x)$. To solve equation (1), the integration is discretized into $j$-1 intervals ( $x_{\text{int}, j}$ $(j = 1, 2, ..., J)$ ) which are much finer than the measured size bins (about 1/50). The activation ratio at $x_{\text{int}, j}$ can be expressed with the activation ratio at measured sizes:

$$A\left(x_{\text{int}, j}\right) = A\left(x_{i(j)}\right) + P_{i(j)}\left(x_{\text{int}, j} - x_{i(j)}\right) \quad (3)$$

$i(j)$ is the ordinal of the measured size closest to $x_{int,j}$. $P_{i(j)}$ can be derived with linear fitting of the 5 activation ratios at sizes close to $x_{int,j}$. Then, equation (1) can be written as

$$R_i = \Delta x_{int} \sum_{j=1}^{J} G(i, x_{int,j}) A(x_{int,j}) n(x_{int,j}) \quad (4)$$

We can set $H_{ij} = \Delta x_{int} G(i, x_{int,j}) n(x_{int,j})$. Then equation (4) can be written as

$$R_i = \sum_{j=1}^{J} H_{ij} A(x_{int,j}) \quad (5)$$

Considering equation (3), equation (5) can be expressed as

$$R_i = \sum_{j=1}^{J} H_{ij} \left[ A(x_{i(j)}) + P_{i(j)} (x_{int,j} - x_{i(j)}) \right]$$

$$= \sum_{j=1}^{J} H_{ij} A(x_{i(j)}) + \sum_{j=1}^{J} H_{ij} P_{i(j)} x_{int,j} - \sum_{j=1}^{J} H_{ij} P_{i(j)} x_{i(j)} \quad (6)$$

$$= \sum_{k=1}^{I} Q_{ik} A(x_k) + \sum_{k=1}^{I} T_{ik} P_k - \sum_{k=1}^{I} Q_{ik} P_k x_k$$

where,

$$Q_{ik} = \sum_{j=1}^{J} H_{ij} \delta(i(j) - k) \quad (7)$$

$$T_{ik} = \sum_{j=1}^{J} H_{ij} x_{int,j} \delta(i(j) - k) \quad (8)$$

$$\delta(x) = \begin{cases} 0, & x \neq 0 \\ 1, & x = 0 \end{cases} \quad (9)$$

We can set $S_i = R_i - \sum_{k=1}^{I} T_{ik} P_k + \sum_{k=1}^{I} Q_{ik} P_k x_k$ which is known. Then equation (6) can be written as

$$S_i = \sum_{k=1}^{I} Q_{ik} A(x_k) \quad (10)$$

Equation set (10) then can be solved by applying nonnegative least squares to minimize $\|S - QA\|$.

**Reviewer:**

*Line 14: Was the SS calibration done with monodisperse particles of known chemical composition?*

**Reply:**

Yes. To make it clear, the sentence has been rewritten as "The SS of CCN counter were calibrated before the campaign and checked at the end of the campaign with monodisperse ammonium sulfate particles (Rose et al., 2008)"

**Reviewer:**

*Line 23: Were the TDMA calibration ammonium sulfate particles monodisperse?*

**Reply:**

Yes. The effective RH of HTDMA was calibrated with monodisperse ammonium sulfate particles automatically every 6 hours. We have revised the corresponding sentence in section 2.4 as "Calibration with monodisperse ammonium sulfate particles was automatically conducted every 6 hours".

**Reviewer:**

*Page 5, Figure 2: The details of the PNSD data that are discussed are not clear to this reader even at 300% magnification due to the high time and size variability of the data. This could perhaps improved by smoothing of the contours. The fine structure in the data is not needed to convey the general features such as "nucleation mode did not start at the lower detection limit of our SMPS", and the beginning and end of particle growth by condensation of vapors. A quantitative label for color scale is needed for panels 1 and 2. The hygroscopic growth at 50 nm is most germane to understanding CCN activation; however -PDF at other diameters would be of interest, perhaps as a supplement to the manuscript.*

**Reply:**

Thanks for the suggestions. We have smoothed the PNSD data before making the contour plot. It looks much clearer now. We have also added the PDF of 50 nm particle shrink factor (at 300 °C) in this figure, as shown in Fig. R2. The labels of color bar for panels of PNSD, $\kappa$-PDF and $f_s$-PDF (PDF of shrink factor at 300 °C) have been added.

[Figure]

Fig. R2 (Fig. 2 in manuscript). 5 NPF events observed during the campaign period. Subplots show the time series of particles number size distribution, $\kappa$-PDF of 50 and $f_s$-PDF of 50 nm particles, and mass fraction of organics and sulfate from PM10 HV-sample analysis.

And we have also added two figures of $\kappa$-PDF and $f_s$-PDF for 50, 100, 150 and 250 nm in supplement (Fig. R2 and R3). We can see the variations of $\kappa$-PDF and $f_s$-PDF for the four sizes are different, since particle with different sizes may have different origin and undergo different aging processes. During NPF events, new particles mainly dominated up to about 80 nm (sometimes also up to 100 nm). The temporal variation of hygroscopicity and volatility of larger particles (150 and 250 nm) may reflect the air mass changes, as can be seen on July 24[th] and 28[th] in Fig. R3.

As answering the general comment of reviewer #2, we agreed that the composition of the new particles may also depend on the air masses, since the air mass routes determine the concentration of precursors to a large extent. In the north NCP, air masses coming from the south are usually more polluted and with high concentration of $SO_2$; while air masses coming from the north are much cleaner (Xu et al., 2011; Ma et al., 2011). During the three MH-type NPF events (July 20[th], 22[nd] and 25[th]), the wind direction was mainly in south section and/or the wind speed was low. The polluted air mass might contain high concentration of $SO_2$, promoting the production of sulfate in particle phase. This might be also the reason of the high stable $\kappa$ of 100 – 250 nm particles on those days. During the two LH-type NPF events (July 24[th] and 28[th]), the wind direction was N/NW and the air mass was clean. The relative contribution of sulfuric acid to the particle growth might be therefore lower. Larger particles also exhibited lower $\kappa$ on the two days, since secondary production might also contribute much on their mass.

[Figure]

Fig. R3 (Fig. S2 in supplement). 5 NPF events observed during the campaign period. Subplots show the time series of particles number size distribution, and $\kappa$-PDF of 50, 100, 150 and 250 nm particles.

[Figure]

Fig. R4 (Fig. S3 in supplement). 5 NPF events observed during the campaign period. Subplots show the time series of particles number size distribution, and $f_s$-PDF of 50, 100, 150 and 250 nm particles.

**Reviewer:**

*Page 6, line 10: The inference of the chemistry of nuclei mode particles from PM10 chemical analysis leaves a lot to be desired. Is there data from previous experiments in the NCP when mass spectrometric analysis was done to aid in the understanding of what the size dependence of sulfate and organic compounds might have been in this field study?*

**Reply:**

We also realized that it was improper to use PM10 chemical composition in our analysis. But we did not find any study which reports the online measurements of chemical composition of sub-100 nm particles in the NCP. Actually our collaborator did AMS measurement during the joint intensive campaign. Unfortunately, we met some problem during transporting of our mobile laboratory. Our measurement therefore started too late and has no overlap with AMS measurement in the campaign. Figure R5 displays the average mass fraction and size-resolved mass distribution of different compounds measured with AMS for the period June 9th to July 9th. We can see organics and sulfate dominates the mass of submicron particles.

Actually, we also did particle volatility measurement with a volatility tandem differential mobility analyzer (TROPOS-type VTDMA; Philippin et al., 2004) during the campaign. The hygroscopicity and volatility of nanoparticles as measured by the HTDMA and VTDMA is often invoked to provide insight into the particle composition (Zhang et al., 2011). We therefore decided to involve this dataset in our study, together with hygroscopicity data, to provide some additional information about the new particles. In the revised manuscript, we built most of the discussion in section 3.1 and 3.2 on the basis of HTDMA

and VTDMA measurements, instead of on PM10 chemical composition data. More details will be show in the response to another comment below (Page 9, line 29 …).

[Figure]

Fig. R5. Average mass fraction (left) and size-resolved mass distribution (right) of different compounds measured with AMS for the period June 9[th] to July 9[th] (this figure is from our collaborator: Key Laboratory for Urban Habitat Environmental Science and Technology, Shenzhen Graduate School of Peking University)

**Reviewer:**

*Line 17: I don't understand "production activity". Do you mean rate of condensation?*

**Reply:**

The sentence has been deleted.

**Reviewer:**

*Line 24: It would be useful in this figure to identify the two events, the sulfate-dominant NPF event and OM-dominant NPF events, either in the caption or under the dates at the top of panel 1.*

**Reply:**

Thanks for the suggestion. The types (MH/LH) of the five NPF events have been added in the title of Fig. 2, 3 and 6 in the manuscript.

**Reviewer:**

*Line 25: "And two types of NPF ... " Does this refer to Yue's work? If yes, then I suggest, "Furthermore, they observed two types of NPF .... "*

**Reply:**

We have deleted this sentence in the revised manuscript.

**Reviewer:**

*Line 31: You present BC mass concentrations. Do you have similar mass concentration data for sulfate and OM? Total sub 800nm and sub 80 nm mass derived from the SMPS volume and an assumed density would be useful as well.*

**Reply:**

Unfortunately, we had no online measurements of sulfate and organic mass concentration during the NPF events.

Following your suggestion, we calculated the mass concentration of sub-80 nm and sub-800 nm, from PNSD and an assumed density 1.6 $gcm^{-3}$. This data has been added in panel B of Fig. 3 and 6 (as shown below). We have also added some description of this data in section 3.2.1 and 3.2.2.

[Figure]

Fig. R6 (Fig. 3 in manuscript). MH-type NPF event on July 22$^{nd}$. Time series of (A) particle number size distribution and geometric mean diameter of nucleation mode, (B) wind speed/direction and the mass concentration of BC, sub-80 nm and sub-800 nm particles, (C) condensational sink (CS) and number concentration of particles in defined size ranges, (D) average $\kappa$ of hygroscopic mode and number fraction of nearly-hydrophobic mode for 50 nm particles, (E) volume fraction remaining of volatile mode and number fraction of non-volatile mode for 50 nm particles, (F) $D_{P,50}$ for 0.20%, 0.40% and 0.80% SS, as well as (G) $S_{50}$ for the three SS. The dashed lines in panel F show the theoretical critical diameters for ammonium sulfate at the three SS. Points filled with color in panel G show the records selected to calculate the average size-resolved activation ratio shown in Fig. 5.

[Figure]

Figure R7 (Fig. 6 in manuscript). LH-type NPF event on July 24[th]. Time series of (A) particle number size distribution and geometric mean diameter of nucleation mode, (B) wind speed/direction and the mass concentration of BC, sub-80 nm and sub-800 nm particles, (C) condensational sink (CS) and number concentration of particles in defined size ranges, (D) average $\kappa$ of hygroscopic mode and number fraction of nearly-hydrophobic mode for 50 nm particles, (E) volume fraction remaining of volatile mode and number fraction of non-volatile mode for 50 nm particles, (F) $D_{P,50}$ for 0.20%, 0.40% and 0.80% SS, as well as (G) $S_{50}$ for the three SS. The dashed lines in panel F show the theoretical critical diameters for ammonium sulfate at the three SS. Points filled with color in panel G show the records selected to calculate the average size-resolved activation ratio shown in Fig. 5.

**Reviewer:**

*Page 7, line 4, Figure 3A The label on the ordinate should be Dp, particle diameter. The color scale label should be placed next to the color scale bar.*

**Reply:**

Thanks for point out this error. It has been corrected. The color scale bar and label for the PNSD contour plot has also been added in Fig. 3 and 6, as shown in Fig. R6 and R7.

**Reviewer:**

*Figure 3 overall: The timing of the appearance of NFP varies depending on the parameter. NFP appear sporadically beginning at 900, Fig 3A. N(40,60) and increases sharply at about 1030. Nuclei mode reaches 50nm about 1200LT. The condensational sink does not decrease sharply over time in the morning. Of course there are other parameters not quantified and presented such as advection or mixing from more polluted layers aloft or actinic flux and photochemical precursor gas formation. These are beyond the scope of the observations and discussion. Simply mention that the indicatory parameters for NFP, Dp,nuclei, N(40,60) and , increase more or less in coincidence over a three hour period.*

**Reply:**

Many thanks for the comment. We fully agreed that the nucleation and growth during this event might be influenced or controlled by some more factors besides condensational sink. To understand the whole process is not the aim of this study, and requires additional observations which are not available. Following your suggestion, we have added "The indicatory parameters for NPF, $N_{[40-60nm]}$ and the geometric mean diameter of nucleation mode, increase in coincidence over a three-hour period" in section 3.2.1.

**Reviewer:**

*Figure 3 D: The legend shows NFH while the right hand label shows NFNH.*

**Reply:**

Thanks for point out this error. It has been corrected in the new figure, as shown in Fig. R6 and R7.

**Reviewer:**

*Line 27, Figure 3E: The highlighting of the points used for calculating the average sizeresolved activation ratio shown in Figure 5 is difficult to see and the change in color is not clear. A larger circle for those points would help the reader at a glance. Also, a theoretical critical Dp line for ammonium sulfate as a reference for the three SS values would be useful for comparison.*

**Reply:**

Thanks for the suggestion. A larger colored circle is now used in panel F in Fig. 3 and 6. And the theoretical critical diameters for ammonium sulfate at the three SS are also marked as dashed lines, as shown in Fig. R6 and R7.

**Reviewer:**

*Page 8, line 30, Figure 5: What is the SS for the campaign average curve? It is difficult to follow all the curves and colors on this plot. I suggest breaking the single plot into three panels, one for each SS. The*

*information in Figure 5 is summarized in Figure 3 panels E and F, but, these do not show the OM vs. sulfate differences or that activation ratios remain below 1.0 even at sizes much larger than Dcrit,SS, so figure 5 is of value for showing this result.*

**Reply:**

Thanks for the suggestion. We have broken Fig. 5 into three panels, one for each SS, as shown in Fig R8. The campaign average activation curves at the three SS are shown as solid black lines in each panel.

[Figure]

Fig. R8: Average size-resolved activation ratio for the selected periods on July 22[nd] and 24[th]

**Reviewer:**

*This last point, AR < 1 at Dp »Dp,crit, needs explanation. Certainly size plays the dominant role as you point out since approximately 80% of the particles activate at Dp50 plus about 5 nm. However, on average only about 85% activation ratio is reached at twice the Dp50. In two cases the maximum AR is only about 60%. Do you have an explanation, e.g., truly insoluble, non-wettable primary particles, or organic surfactants?*

**Reply:**

We thought this relatively low activation ratio at sizes much larger than $D_{p,50}$ was caused by the externally mixed BC particles which is completely hydrophobic, because 1) emission of BC particles from fossil fuel and bio fuel consuming usually increased in the evening due to increased traffic and cooking, 2) the vertical mixing and photochemical aging process were inhibited in the nocturnal boundary layer. It can be seen in Fig. R9 that the number fraction of non-volatile and nearly-hydrophobic particles had very similar variation as the BC mass concentration in the afternoon and evening, especially for 100 and 150 nm.

July 24[th] is actually a kind of extremely case of BC influence. We can see the BC mass concentration in the evening of July 24[th] is similar as that in the evening of July 22[nd]. However, the air mass was very clean in the daytime of July 24[th] (can be seen also from the $m_{[9-800nm]}$ in panel B of Fig. R6 and R7). Due to the low concentration of background aerosol particles, the relative contribution of BC in the evening was higher than in the other days. Thus the number fraction of externally mixed BC particles is higher, resulting in a low activation ratio (Fig. R8) and high number fraction of non-volatile and nearly hydrophobic mode (Fig. R9). The decrease of the activation ratio at size range of 100 – 200 nm for 0.80% and 0.40% SS is probably because the number fraction of externally mixed BC particles at 150 nm is higher than at 100 nm, which is also reflected in $NF_{NV}$ and $NF_{NH}$ (Fig. R9).

To make it clear, we have added "And the activation ratio reaches only about 80% even at the size of $D_{P,50} \times 2$, which is probably due to the high concentration of externally mixed BC particles (also shown as a clear near-hydrophobic and non-volatile mode at 100 and 150 nm in Fig. S2 and S3)." in section 3.2.1, and "The increasing anthropogenic emission caused a decrease in particle hygroscopicity and CCN activity (Figure 6D, F and G). However, the activation ratio at size range of 100 – 200 nm is much lower than that in the nighttime of July 22[nd] (Fig. 5). This is probably because the concentration of background aerosol particles was lower on July 24[th] (also can be seen from the $m_{[9-800nm]}$ in panel B of Fig. 3 and 6). The relative contribution of BC particles in the evening was therefore higher, resulting in a high number fraction of externally mixed BC particles." in section 3.2.2.

[Figure]

Fig. R9. Temporal variation of BC mass concentration, $NF_{NH}$ and $NF_{NV}$ on July 22[nd] and July 24[th]

**Reviewer:**

*Page 8, line 11, Figure 6: Change caption and legends and plotting in figure 6 as suggested for figure 3. See Word changes in figure 3 caption.*

**Reply:**

Figure 6 has been improved in the same way as Fig. 3.

**Reviewer:**

*Page 9, line 29: Do you know from chemical measurement that sulfate dominated the particle growth size mode, i.e., nuclei and Aitken modes? From PM10 sulfate analysis alone that cannot be supported. The best evidence you have is for sulfate vs. OM is from your and NFNH.*

**Reply:**

Many thanks for this comment.

We have also realized that it was improper to use PM10 chemical composition in our analysis. Most of the OM and sulfate mass is in accumulation mode which is dominated not only by secondary production but also by horizontal and vertical transportation. The mass fractions of sulfate and OM in nucleation mode are therefore not necessarily proportional to their mass fractions in PM10. Unfortunately, we have no measurement about size-resolved particle chemical composition in ultrafine size range.

However, we did particle volatility measurement with a volatility tandem differential mobility analyzer (TROPOS-type VTDMA; Philippin et al., 2004) during the campaign. The hygroscopicity and volatility of nanoparticles as measured by the HTDMA and VTDMA is often invoked to provide insight into the particle composition (Zhang et al., 2011). We therefore decided to involve this dataset in our study, together with hygroscopicity data, to provide some additional information about the new particles.

Non-volatile cores were found in 50 nm newly formed particles in all the five NPF events (Fig. R6 and R7). This phenomenon was also observed in Melpitz, Germany (Wehner et al., 2005) and Hyytiälä, Finland (Ehn et al., 2007). The sizes of the non-volatile residues of 50 nm new particles were however different in the five NPF events. In the NPF event on July 20th, 22nd and 25th, the majority of 50 nm new particles exhibited a shrink factor of about 0.3; while on July 24th and 28th, the shrink factor is a slightly higher, about 0.4. It means more polymer-type organics were formed during the growth of new particles on July 24th and 28th. In another study in the same region, Yue et al. (2010) found that sulfuric acid was the major contributor of the growth of newly formed particles, and organic compounds might also play a major role in some cases. Combining all the information above and the result of particle hygroscopicity, it is very likely that condensation and neutralization of sulfuric acid dominated the growth of the new particles on July 20th, 22nd and 25th; while its contribution was much lower on July 24th and 28th and organic compounds had an important contribution to particle growth. However, since we have no direct measurement of nanoparticle chemical composition, to be more accurate, we have modified the name of the two types of NPF events. The events on July 20th, 22nd and 25th were termed MH-type NPF (NPF with More Hygroscopic particles); while the events on July 24th and 28th were termed LH-type NPF (NPF with Less Hygroscopic particles).

Our study aimed at evaluating the variability of the CCN activity during NPF events in an anthropogenically polluted atmosphere and the applicability of some simplified CCN parameterization. Although the chemical composition measurement of ultrafine particles was not available, the online measurements of hygroscopicity and volatility can provide some indication for distinguishing different NPF types. In the revised manuscript, we built most of the discussion in section 3.1 and 3.2 on the basis of HTDMA and VTDMA measurements, instead of on PM10 chemical composition data.

We have deleted most of the content about PM10 composition data, added a new section to introduce VTDMA measurement, revised section 3.1 and 3.2, added VTDMA data in figure 2, 3 and 6 (shown as Fig. R2, R6 and R7), and replaced figure 4 with a new figure of average $\kappa$-PDF and $f_s$-PDF during NPF (Fig. R10). In the following we list the major new content which has been added in the revised manuscript (in dark blue color).

In section 2.5:

[revised manuscript text omitted]

**Reviewer:**

*Page 10, lines 1 through 3: It is not clear how the difference ratio plotted in Figure 7 was calculated, what NCCN ref and NCCN Aravg are. The text and figure caption and x-scale label symbols are not consistent. Provide an equation (4) in the text for the bias parameter used in the figure e.g. CCNbias = ..........*

**Reply:**

Thanks for the comment.

$N_{CCN,AR\text{-}ave}$, $N_{CCN,D\text{-}ave}$ and $N_{CCN,ref}$ are defined before equation (3): "To answer this question, the CCN number concentration was calculated with eq. (2) based on the campaign average activation ratio shown as the solid black line in panel A of Fig. 5 (the calculated CCN number concentration is termed $N_{CCN,AR\text{-}ave}$), campaign average critical diameter (termed $N_{CCN,D\text{-}ave}$) and real-time activation ratio (termed $N_{CCN,ref}$) at 0.80% SS."

We have added two equations to define the relative difference plotted in Fig. 7: "The relative difference between $N_{CCN,AR\text{-}ave}$ and $N_{CCN,ref}$ (termed CCNbiasA), and between $N_{CCN,D\text{-}ave}$ and $N_{CCN,ref}$ (termed CCNbiasB), was then evaluated. CCNbiasA and CCNbiasB were respectively calculated as

$$CCNbiasA = \frac{N_{CCN,AR-ave} - N_{CCN,ref}}{N_{CCN,ref}} \qquad (4)$$

$$CCNbiasB = \frac{N_{CCN,D-ave} - N_{CCN,ref}}{N_{CCN,ref}} \qquad (5)"$$

And the caption of Fig. 7 has been revised.

**Reviewer:**

*Line 13, Figure 7: The caption should the time period of average, i.e., afternoons of July 22nd and July 24th. I assume the frequency statistics are number of occurrences per interval of relative ratio, and for the 2 dim, number of occurrences per interval of relative ratio and time. Mention this in the caption.*

**Reply:**

Fig. 7 in the manuscript is based on the data of the entire campaign period, not only the two NPF days. The caption of Fig. 7 has been revised as "Fig. 7. 2 dimensional frequency distribution (number of occurrences per interval of CCNbias and time of day, shown as contour plot) and overall frequency distribution (number of occurrences per interval of CCNbias, shown as red line) of CCNbiasA (panel A) and CCNbiasB (panel B) for the entire campaign period"

**Reviewer:**

*Are there significant differences between the two parameters? To me they look very similar other than the left hand tail. You do not mention the chemistry or physics behind this tail. Nor can I imagine any.*

**Reply:**

The frequency distributions of CCNbiasA and CCBbiasB are very similar since the average activation ratio curve and average critical dimeter used in the calculation are derived from the same dataset which was also used this sensitivity test. In other words, the average activation ratio curve and average critical diameter well represented the CCN activity during the campaign period. Therefore, the majority of the CCNbiasA and CCNbiasB are located around 0, within ±10%.

The left hand tails of the two distributions represent the data during NPF events. The campaign average activation ratio and critical diameter were not appropriate for $N_{CCN}$ prediction during NPF events (especially during MH-type). Using average critical diameter may cause larger negative bias in $N_{CCN}$. Figure R11 shows the measured AR curve during the NPF event on July $22^{nd}$, and the campaign average AR curve and the critical diameter. Compared with the real-time AR curve (red line), using average critical diameter (a stepwise size-resolved activation ratio, blue line) overestimates $N_{CCN}$ at $D_P > D_{P,cri}$ and underestimates $N_{CCN}$ at $D_P < D_{P,cri}$. If PNSD and activation ratio curve are similar as the average ones (e.g. on non-NPF days), those overestimation and underestimation may compensate. However, during NPF events, the activation ratio curve may shift towards lower size (in MH-type NPF) and particle number concentration in ultrafine size range increases significantly. The underestimation in the left side of $D_{P,cri}$ is therefore much higher than the overestimation in the right side of $D_{P,cri}$, and resulting in a large negative CCNbiasB.

We had added some discussion about this in section 3.3: "This indicates that to use an average $D_{P,cri}$ may result in a larger underestimation of calculated $N_{CCN}$ during NPF events in the NCP. This is because using such a stepwise size-resolved activation ratio overestimates $N_{CCN}$ at $D_P > D_{P,cri}$ and underestimates $N_{CCN}$ at $D_P < D_{P,cri}$. If real-time PNSD and activation ratio curve are similar as the average ones, those overestimation and underestimation may compensate. However, during NPF events, the activation ratio curve may shift towards lower size (in MH-type NPF) and particle number concentration in ultrafine size

range increases significantly. The underestimation of $N_{CCN}$ in the left side of $D_{P,cri}$ is therefore much higher than the overestimation in the right side of $D_{P,cri}$, and resulting in a large negative CCNbiasB."

[Figure]

Fig. R11: measured AR curve during the NPF event on July 22$^{nd}$, and the campaign average AR curve and the critical diameter

**Reviewer:**

*In other plots and discussions you have presented and compared data from 22 July vs. 24 July. Were there significant differences in the frequency distribution plots for those two days?*

**Reply:**

Fig. R12 shows the frequency distribution (number of occurrences per interval of CCNbias) of CCNbiasA (left panel) and CCNbiasB (right panel) for July 22$^{nd}$ and July 24$^{th}$.

Calculated with campaign average AR curve, CCNbiasA has much more negative values (left tail) on July 22$^{nd}$ than on July 24$^{th}$, due to the occurrence of MH-type NPF event on July 22$^{nd}$. Calculated with campaign average critical diameter, CCNbiasB on both days shows big left tail in their frequency distributions. The number of negative values on July 22$^{nd}$ is also higher than that on July 24$^{th}$.

Probably because NPF usually happens in an atmosphere which is not very polluted (low condensational sink), the influence of freshly emitted BC on CCN activity is quite large in the evening of the two NPF days. Positive CCNbias up to +30% can be seen on both days.

[Figure]

Fig. R12. Frequency distribution (number of occurrences per interval of CCNbias) of CCNbiasA (left panel) and CCNbiasB (right panel) on July 22$^{nd}$ and July 24$^{th}$.

**Reviewer:**

*Page 11, line 31: " ... bias of the estimated NCCN ranges from about 0 to 30% ... " From your frequency distribution plots in Figure 7, I would put the biases at 10%. That is my simple ocular analysis. Better would be the percent bias at one or two standard deviations of the of the data.*

**Reply:**

The frequency distribution plot in Fig. 7 in the manuscript is for the entire campaign measurement. Therefore most of the bias values are within ±10%. The bias during NPF events is usually larger, especially during MH-type NPF (shown as the left tail of the frequency distribution in Fig. 7). To avoid confusion, the sentence has been revised as "From the discussion in section 3.3 we learned that the bias of the estimated $N_{CCN}$ during NPF events can be up to 30% if a fixed activation ratio curve (best estimation, representative of the region and season) is used"

**Reviewer:**

*Page 12, line 25: I would make a stronger statement. This means that during NPF in the NCP, the CCN activity of newly formed particles was different during these two events; the aerosol present during the organic dominated event was less hygroscopic and less active as CCN for a given diameter and supersaturation.*

**Reply:**

Thanks for the suggestion. The sentence has been revised as "This means that during NPF in the NCP, the CCN activity of newly formed particles was different during these two events; the aerosol present during the LH-type event was less active as CCN for a given diameter and supersaturation"

**Reviewer:**

*Line 29: Again your 0 to 30% values could be more accurately presented as I suggested above, page 11, line 31.*

**Reply:**

Thanks for the suggestion. Similar as for P11L31, the sentence has been revised as "
[revised manuscript text omitted]
 As the newly formed particles grow grew up to 50 nm and become dominated the majoritynuclei mode number, the number fraction of hydrophobic mode particles (NF$_{NH}$) decreases decreased from about 0.2 to 0. Such a great variationlarge decrease in particle hygroscopicity indicates that the chemical composition of 50 nm particles during the NPF event is was different from that of pre-existing ones.

The mass concentration and fraction of the major compounds in PM10 is shown in Figure 4. As mentioned in section 3.1, the ultrafine particles were only take a minor fraction in of PM10 total mass. However, the temporal variation of the mass fraction of different chemical compounds may provide some information about the secondary aerosol production. We can see that SO$_4^{2-}$ takes was 21.5% of PM10 total mass concentration on average during the daytime of July 22nd (11:00 LT

17:52 LT), which is almost double of that for of the night before (July 21[st] 18:55 LT – July 22[nd] 06:55 LT). This indicates that the secondary production of sulfate is was very active during that period, and makes constituted a major contribution on to the PM10 mass. Considering the high hygroscopicity of the newly formed particles (average $\kappa$ about 0.45), it is reasonable to assume that sulfate takes constitutes the main mass fraction in those particles.

During this NPF event, an enhancement of aerosol CCN activity can be seen. As the $N_{[40-60nm]}$ increases increased sharply at around 10:30 LT, $D_{P,50}$ decreases decreased from about 46 nm to 39 nm for 0.80% SS, and decreased from about 70 nm to 60 nm for 0.40% SS (Fig.ure 3FE). The $S_{50}$ increases increased from about 3.5 to 6.0 for 0.80% SS and increases increased from about 4 to 6 for 0.40% SS (Fig.ure 3GF), meaning that the size-resolved activation ratio curve gets became steeper. It is interesting that enhancement of aerosol CCN activity can also be seen for 0.20% SS, for which $D_{P,50}$ is out oflarger than the size range dominated by the newly formed particles (Fig.ure 3C). This is because the secondarily produced low volatility, water soluble compounds produced by gas phase reactions may condense on all particles. The CCN activity of pre-existing particles might therefore also increase.

To have a better view of particle CCN activity during the NPF event of July 22[nd], three records of size-resolved activation ratio before and during the NPF event are selected and averaged (the corresponding records are marked as color-filled points in Fig.ure 3FE), as shown in Fig.ure 5. The activation ratio before the nucleation is basically the same as the campaign average for at all the three SS. However, the activation ratio curves obviously shifted towards lower size and get became steeper during the NPF event for SS of 0.40% and 0.80%, indicating that the particles are were more hygroscopic, and with had a narrower probability distribution of hygroscopicity compared with the pre-existing particles (Su et al., 2010).

In the nighttime after 18:00 LT, due to the collapse of the boundary layer and the increase of aerosol emission (traffic and cooking), the influence of anthropogenic emission starts to be visible in the time series of particle number size distribution (Fig.ure 3A). The newly formed particles keep on growinggrew further largely contributed by the through coagulation and condensation of the freshly emitted particulate and gaseous pollutants, and therefore become became less hygroscopic. The BC mass concentration increases increased significantly after 18:00 LT and shows a peaked of at about 20 μgm[-3] at around 22:00 LT (Fig.ure 3B), resulting in an increase of the number fraction of hydrophobic mode particles (Fig.ure 3D and S2). These results indicate that in the nighttime the a major fraction of the particle population is getting became less hygroscopic, and that different compounds are 
[revised manuscript text omitted]

---

## Author Comment (AC2) · 23 May 2016

**Response to comments of referee #2**

**General comments**

*This paper investigated the ability of aerosols acting as CCN during the new particle formation (NPF) events over North China Plain. The authors focused on two different NPF events and suggested that the possible deviation of simplified NCCN estimation might be significant for the NPF periods.*

*Overall comment: The authors differentiated two cased based on the sulfate fraction in PM10 to conclude the sulfate dominant and OM dominated cases. However, the sulfate fraction in PM10 cannot be representative for sub-micron sized particles and that was also stated by the authors. If sulfate fraction in PM10 can represents for the sulfate fraction in sub-micrometer sized particles, the authors should also expect the high sulfate fraction should correspond to higher kappa for all periods and this is not always true. For example, the kappa of 7/20 in the early morning showed a much lower value even though the sulfate fraction is similar to the daytime of 7/20. This analytical method might be suspicious without further validation. The composition variation might be due to the parcel source since the provided two cases has the source difference; i.e. west-southwest wind for 7/22 and northwest wind for 7/24. Could the radiation also play a role affecting the hygroscopicity because 7/22 is a hazy day while 7/24 is a sunny day? Overall, it is difficult to conclude the exact factor with only two cases.*

**Reply:**

Many thanks for this comment.

We have also realized that it was improper to use PM10 chemical composition in our analysis. Most of the OM and sulfate mass is in accumulation mode which is dominated not only by secondary production but also by horizontal and vertical transportation. The mass fractions of sulfate and OM in nucleation mode are therefore not necessarily proportional to their mass fractions in PM10. Unfortunately, we have no measurement about size-resolved particle chemical composition in ultrafine size range. Actually, our collaborator did AMS measurement during the joint intensive campaign. But we met some problem during transporting of our mobile laboratory. Our measurement therefore started too late and has no overlap with AMS measurement in the campaign.

However, we did particle volatility measurement with a volatility tandem differential mobility analyzer (TROPOS-type VTDMA; Philippin et al., 2004) during the campaign. The hygroscopicity and volatility of nanoparticles as measured by the HTDMA and VTDMA is often invoked to provide insight into the particle composition (Zhang et al., 2011). We therefore decided to involve this dataset in our study, together with hygroscopicity data, to provide some additional information about the new particles.

Non-volatile cores were found in 50 nm newly formed particles in all the five NPF events (Fig. R1 and R2). This phenomenon was also observed in Melpitz, Germany (Wehner et al., 2005) and Hyytiälä, Finland (Ehn et al., 2007). The sizes of the non-volatile residues of 50 nm new particles were however different in the five NPF events. In the NPF event on July 20[th], 22[nd] and 25[th], the majority of 50 nm new particles exhibited a shrink factor of about 0.3; while on July 24[th] and 28[th], the shrink factor is a slightly higher, about 0.4. It means more polymer-type organics were formed during the growth of new particles on July 24[th] and 28[th]. In another study in the same region, Yue et al. (2010) found that sulfuric acid was

the major contributor of the growth of newly formed particles, and organic compounds might also play a major role in some cases. Combining all the information above and the result of particle hygroscopicity, it is very likely that condensation and neutralization of sulfuric acid dominated the growth of the new particles on July 20[th], 22[nd] and 25[th]; while its contribution was much lower on July 24[th] and 28[th] and organic compounds had an important contribution to particle growth.

The possible difference in the chemical composition of new particles might be cause by different factors, e.g. the concentration of precursors and the activity of some reactions. These factors might be determined or influenced by other parameters, e.g. ambient temperature, radiation, air mass origin, and vertical mixing. We agreed that the composition of the new particles may strongly depend on the route of air masses, since the air mass routes determine the concentration of precursors to a large extent. In the northern NCP, air masses coming from the south are usually more polluted and with high concentration of $SO_2$; while air masses coming from the north are much cleaner (Xu et al., 2011; Ma et al., 2011). During the three MH-type NPF events (July 20[th], 22[nd] and 25[th]), the wind direction was mainly in south section and/or the wind speed was low. The polluted air mass might contain high concentration of $SO_2$, promoting the production of sulfate in particle phase. During the two LH-type NPF events (July 24[th] and 28[th]), the wind direction was N/NW and the air mass was clean. The relative contribution of organic compounds to the particle growth might be therefore higher. The radiation might also play some roles, but we have no radiation measurements during the campaign.

In this study, we aimed at evaluating the variability of the CCN activity during NPF events in an anthropogenically polluted atmosphere and the applicability of some simplified CCN parameterization. To find out the reason behind the composition variation of the new particles is out of the scope of this study, and needs more additional measurements which are not available. So we decided to focus on CCN activity and not go deep into questions about particle composition. Although the chemical composition measurement of ultrafine particles was not available, the online measurements of hygroscopicity and volatility can already provide information for distinguishing different NPF types and explaining the variation in CCN activity.

In the manuscript, we now build most of the discussion in section 3.1 and 3.2 on the basis of HTDMA and VTDMA measurements, instead of on PM10 chemical composition data. Since we have no direct measurement of nanoparticle chemical composition, to be more accurate, we have modified the name of the two types of NPF events. The events on July 20[th], 22[nd] and 25[th] were termed MH-type NPF (NPF with More Hygroscopic particles); while the events on July 24[th] and 28[th] were termed LH-type NPF (NPF with Less Hygroscopic particles).

We have deleted most of the content about PM10 composition data, added a new section to introduce VTDMA measurement, revised section 3.1 and 3.2, added VTDMA data in figure 2, 3 and 6 (shown as Fig. R1, R2 and R3), and replaced figure 4 with a new figure of average $\kappa$-PDF and $f_s$-PDF during NPF (Fig. R4). In the following we list the major new content which has been added in the revised manuscript (in dark blue color).

In section 2.5:

[revised manuscript text omitted]

**Other comments**

**Reviewer:**

*1. Lines 4-11 of Page 4: How many sizes did the authors scan for a given SS?*

**Reply:**

The DMA-CCNC system is operated in a size-scanning mode. After the data inversion, the final activation ratio curve was given at 40 sizes, from about 9 to 300 nm. To make it clear, we added "The system is operated in a size-scanning mode" into the text.

**Reviewer:**

*2. There were two methods applied to determine the hygroscopicity. How is the comparison?*

**Reply:**

The $\kappa$ derived from 50% activation ratio ($D_{P,50}$) from DMA-CCNC measurement, and derived from hygroscopic growth factor measured with HTDMA are shown in Fig. R5. Generally, $\kappa$ derived with two measurements agree with each other. For larger sizes, CCNC-derived $\kappa$ are higher than HTDMA-derived $\kappa$. This is probably because the $\kappa$ of some secondary organic aerosols (SOA) may increase at high RH. SOA-particles were more CCN active than suggested by their sub-saturated growth factors at RH<98% (Wex et al., 2009 and references therein). We also added Fig. R5 into the supplement as a reference (Fig. S1).

[Figure]

Fig. R5. Comparison of $\kappa$ derived with DMA-CCNC measurement and HTDMA measurement.

**Reviewer:**

*3. Line 20 of Page 20: the authors stated there are 10 out of 28 days associated with NPF events but only 5 days with particle growing. What parameter controls the growing process? Does the hygroscopicity of aerosols at these two cases different?*

**Reply:**

Thanks for this comment. We realize that the statement "NPF events were observed on 10 of 28 days and clear growth of newly formed particles was observed on 5 of those days" in the manuscript is somehow inaccurate. The 10 observed NPF events during the campaign are shown in Fig. R6, R7 and R8. Actually, the 5 events shown in Fig. 2 in the manuscript are good NPF cases with very clear growth and without strong fluctuation in diameter and concentration of the nucleation mode (i.e. Class I in the classification system in Dal Maso et al., 2005). In the other 5 events, we also observed particle growth in 4 of them, which is weak and/or with strong fluctuations (i.e. Class II and undefined events in the classification system in Dal Maso et al., 2005) (Fig. R7). There is only one event which is clearly without growth (Fig. R8).

To make it clearer, we revised the statement in section 3.1 as "During our intensive field campaign, NPF events were observed on 10 of 28 days. Clear growth of nucleation mode particles without strong fluctuations in diameter or concentration (i.e. Class I of the classification system in Dal Maso et al., 2005) was observed on 5 of those days."

There are many possible parameters which may influence or interrupt the growth of new particles, for example, variations in solar radiation (cloud cover), temperature, and concentration of precursors. Moreover, NPF usually occurred inhomogeneously in horizontal and vertical scale in a region. Horizontal wind and vertical mixing may also result in fluctuations in the PNSD measured at a fixed point. As an example, we did not observe the growth of nucleation mode particles on July 16[th], probably because the strong north wind blew the newly formed particles away from the area of our station (Fig. R8). With the measurement we have, it is difficult to evaluate the parameter(s) controlling the growth process.

The hygroscopicity and volatility of 50 nm particles during the 10 NPF events are also displayed in Fig. R6, R7 and R8. On August 8[th], when new particles dominated 50 nm (around 14:00 LT), we can see a decrease in both $\kappa_{ave,H}$ and $VFR_V$. The values of $\kappa_{ave,H}$ and $VFR_V$ are similar as those in the other 2 LH-type NPF (July 24[th] and 28[th]), so this event can be also classified as a LH-type event. For the other 4 events (July 16[th], 17[th], 21[st] and 23[rd]), since 50 nm was not dominated by the new particles and/or some data is missing due to instrumental issues, it is difficult to make any conclusion on the hygroscopicity and volatility of the new particles.

[Figure]

Fig. R6. Wind speed and direction, BC mass concentration, particle number size distribution, average $\kappa$ of hygroscopic mode and number fraction of hydrophobic mode for 50 nm particles, volume fraction remaining of volatile mode and number fraction of non-volatile mode for 50 nm particles of the 5 NPF events with clear and smooth growth (i.e. Class I in the classification system in Dal Maso et al., 2005).

[Figure]

Fig. R7. Wind speed and direction, BC mass concentration, particle number size distribution, average $\kappa$ of hygroscopic mode and number fraction of hydrophobic mode for 50 nm particles, volume fraction remaining of volatile mode and number fraction of non-volatile mode for 50 nm particles of the 4 NPF events with unclear growth (i.e. Class II and undefined events in the classification system in Dal Maso et al., 2005).

[Figure]

Fig. R8. Wind speed and direction, BC mass concentration, particle number size distribution, average $\kappa$ of hygroscopic mode and number fraction of hydrophobic mode for 50 nm particles, volume fraction remaining of volatile mode and number fraction of non-volatile mode for 50 nm particles of the NPF events without growth.

**Reviewer:**

*4. Because the authors focused on the NPF events, the date of Figure 4 should be the same as Figure 2.*

**Reply:**

Figure 4 in manuscript has been replaced with a new figure of average $\kappa$-PDF and $f_s$-PDF of 50 nm particles during NPF, as shown as Fig. R4.

**Reviewer:**

*5. Line 10 of Page 9: What is the average activation ratio profile the authors applied?*

**Reply:**

The average activation ratio curve for 0.80% SS (shown as solid black line in panel A of Fig. 5 in manuscript) was applied in the calculation here. To make it clear, the sentence has been revised as "To answer this question, CCN number concentration was respectively calculated with Eq. (2) based on the campaign average activation ratio (shown as the solid black line in Fig. 5A; the calculated CCN number concentration is termed $N_{CCN,AR\text{-}ave}$), campaign average critical diameter (termed $N_{CCN,D\text{-}ave}$) and real-time activation ratio (termed $N_{CCN,ref}$) at 0.80% SS."

**Reviewer:**

*6. The label of y axis for all SMPS figures should be Diameter (nm) not the number concentration.*

**Reply:**

Thanks for point out the mistake. They have been corrected, as shown in Fig. R1, R2 and R3.

**Reviewer:**

*7. The color bar for SMPS and k in Figure 2 should be provided.*

**Reply:**

The colorbar for SMPS contour plot has been added, as shown in Fig. R3.

**Reviewer:**

*8. There are two ticks for the right y axes of Figure 6B, 6C and 6D which should be modified.*

**Reply:**

Thanks for the comment. The wrong ticks have been removed, as shown in Fig. R1 and R2.

case study, Tthe NPF events on July 22nd (MH-type NPF) and 24th (LH-type NPF) are selected and discussed in detail in this section. Considering the variation of the particle hygroscopicity and the mass fraction of sulfate and OM, we name identify these two events as sulfate-dominant NPF event and OM-dominant NPF events, respectively..

**3.2.1 CCN activity in MH-type in sulfate-dominant NPF eventNPF**

Figure 3 illustrates details of the NPF event that occurred on July 22nd. July 22nd is was a hazy day with the average temperature, RH and wind speed of 26.1 °C, 77.5% and 0.4 ms$^{-1}$, respectively. The weak south-southwest wind starting continuing from the previous day facilitateds the accumulation of pollutants in the region (Xu et al., 2011). The daily average BC mass concentration is was 6.94 μgm$^{-3}$, 50% higher than the average 4.66 μgm$^{-3}$ of the entire measurement period (4.66 μgm$^{-3}$).

As the developing of the boundary layer developed in the morning, the the BC mass concentration and the particulate particle condensational sink (CS, Kulmala et al., 2001) and the mass concentration of BC and sub-800 nm particles started to decrease at 07:00 LT. The newly formed particles started to be visible in our SMPS record at around 09:00 LT. The indicatory parameters for NPF, $N_{[40-60nm]}$ and the geometric mean diameter of nucleation mode, increase in coincidence over a three-hour period. Large amount of secondary particulate matter was produced during the growth of new particles, shown as a sharp increase in sub-80 nm particle mass concentration.; and The new particles keep oncontinued growing until the end of the day with an average growth rate of 6.3 nm h$^{-1}$ (Figure Fig. 3A). The newly formed particles reached 50 nm at around 10:30 LT, resulting in a sharp elevation of $N_{[40-60nm]}$ from about 2×10$^3$ cm$^{-3}$ to 1×10$^4$ cm$^{-3}$. Correspondingly, the number fraction of nearly hydrophobic mode particles ($NF_{NH}$) and non-volatile mode particles ($NF_{NV}$) decreased from about 0.2 to 0, meaning that the newly formed particles grew to 50 nm and dominated the nuclei mode number. This can also be confirmed by the average $\kappa$-PDF and $f_\kappa$-PDF during the NPF event (Fig. 4). Both $\kappa$-PDF and $f_\kappa$-PDF exhibited narrow unimodal patterns, indicating that the majority of 50 nm particles originated from the same source, NPF. Identifying as MH type, during the NPF event, the 50 nm new particles exhibited a much higher $\kappa_{ave,H}$ than the pre-existing particles (about 0.45 vs. 0.3). As discussed in section 3.1, it is very likely that the condensation and neutralization of sulfuric acid contributed most to the growth of the new particles.

Correspondingly, the average $\kappa$ of hygroscopic mode particles ($\kappa_{ave,H}$) increased from around 0.3 to 0.45 within 1 hour (Figure 3D). Since As the newly formed particles grow grew up to 50 nm and become dominated the majoritynuclei mode number, 
[revised manuscript text omitted]

---

## Referee Report (RR1)

The authors have done an extensive revision which addresses, corrects, explains the many comments of the reviewers and the satisfaction of this reviewer.

A few, more minor points follow.

Cheng, Y. F., et. al. 2009, addressed the effect of aging on aerosol optical properties. What might the effect of ageing be on the CCN activity distributions? I'm thinking of the importance of activation as a function of size, and NPF and chemical and physical processes on a regional (or larger scale) and how your results apply to global models of cloud processes. Admittedly your data set is small and as you emphasized limited. Thus, this question may be premature.

Page 2 line 28
... extend to diameters much larger ...

Page 5 line 15
Shrink factor. Better would be "shrinkage factor" , (cf. growth factor0.
Or: "Thermal volatility shrinkage factor" the first time it is mentioned  (cf. hygroscopic growth factor).
(Shrink being a verb is not proper;  shrinkage, like growth, being an noun that can be used as an adjective, is better.  Your picky English pedant.)

Page 6 line 32 and in conclusion
.... formed particles is largely different in the five events.

Were these significantly different either by statistical analysis or ocular analysis?

Page 8 line 21
**A** large amount of

Page 9 line 28
This day was relatively clean with **a** cloudless blue sky.

line 30
(1.74 μgm-3) **during**  the daytime due to

line 34
As the newly formed particles grew to 50 nm and **became** the

Page 10 line 8
meaning that **more of** the  polymer-type organics **were** produced during the growth of the new particles in this event.

page 12 line 1 and following

**Negative**  biases, ranging from

line 10
majority of the relative bias **is** still locate**d** between ....

line 34
 **a consequence of** growth processes

---

## Author Response (AR2)

Dear Editor,

A point-by-point response to the comments of referee #1 and a revised manuscript were uploaded. We would like to thank the two referees again for their helpful comments.
And also thank you very much for editing our manuscript!

Best Regards
Nan Ma and co-authors

**Response to comments of referee #1**

**Reviewer:**

*The authors have done an extensive revision which addresses, corrects, explains the many comments of the reviewers and the satisfaction of this reviewer. A few, more minor points follow.*

*Cheng, Y. F., et. al. 2009, addressed the effect of aging on aerosol optical properties. What might the effect of ageing be on the CCN activity distributions? I'm thinking of the importance of activation as a function of size, and NPF and chemical and physical processes on a regional (or larger scale) and how your results apply to global models of cloud processes. Admittedly your data set is small and as you emphasized limited. Thus, this question may be premature.*

**Reply:**

Many thanks for the comments.

Our study actually partly answered the question raised by the reviewer ("What might the effect of ageing be on the CCN activity distributions?"), since the growth of the newly formed particles is also a kind of aerosol aging. As discussed in the manuscript, the chemical composition of new particles is almost entirely determined by the condensing material. Therefore, CCN activity of those particles strongly depends on the aging process.

More generally, I think the effect of the aerosol aging on its CCN activity might be complex, depending on the sizes and types of aerosol particles, concentrations of precursors, and ambient condition parameters, etc. I think it is very worthy investigating the effect of aging on particle CCN activity, especially for large particles which are easier to become a CCN. And the North China Plain provides us a good platform and opportunity for preforming such kind of studies, since the concentration of precursors are high and particles may get aged very fast (Guo et al., 2014). We will try to study the effect of aging on CCN activity of larger particles based on our dataset, and will prepare another manuscript if we got any interesting results.

As pointed out by the reviewer, to improve the cloud processes in global models needs much larger datasets covering longer time periods and different types of region. Our results might be only applicable in regional model studies for the NCP. However, our study emphasizes that to evaluate the contribution of new particle formation on CCN number concentration, one should be careful on the CCN activity of ultrafine particles, since it may largely differs in different cases.

**Reviewer:**

*Page 2 line 28 ... extend to diameters much larger ...*

**Reply:**

Thanks. It has been corrected.

**Reviewer:**

*Page 5 line 15 Shrink factor. Better would be "shrinkage factor" , (cf. growth factor0. Or: "Thermal volatility shrinkage factor" the first time it is mentioned (cf. hygroscopic growth factor).*

*(Shrink being a verb is not proper; shrinkage, like growth, being an noun that can be used as an adjective, is better. Your picky English pedant.)*

**Reply:**

Thanks for the suggestion. "Thermal volatility shrinkage factor" and "shrinkage factor" are now used in the manuscript instead of "shrink factor"

**Reviewer:**

*Page 6 line 32 and in conclusion .... formed particles is largely different in the five events.*

*Were these significantly different either by statistical analysis or ocular analysis?*

**Reply:**

I am sorry that this sentence might cause some confusion.

I did not mean that the newly formed particles in NPF event have a broad $\kappa$-PDF. Actually, the $\kappa$-PDF of 50 nm particles during NPF event is quite narrow (Fig. 4 in manuscript). What I wanted to discrib is that the newly formed particles in different NPF events exhibit different level of $\kappa$. To avoid possible confusion, the sentence in P6L32 has been revised as "It is interesting to see in Fig. 2 that 50 nm particles exhibited different levels of $\kappa$ in the five NPF events". And the sentence if conclusion has been revised as "In the 5 NPF events, newly formed particles exhibited different hygroscopicity and volatility, suggesting that the particle growth might be dominated by different species during different NPF events".

**Reviewer:**

*Page 8 line 21*

*A large amount of*

**Reply:**

Thanks. It has been corrected.

**Reviewer:**

*Page 9 line 28*

*This day was relatively clean with **a** cloudless blue sky.*

**Reply:**

Thanks. It has been corrected.

**Reviewer:**

*line 30*

*(1.74 μgm-3) **during**  the daytime due to*

**Reply:**

Thanks. It has been corrected.

**Reviewer:**

*line 34*

*As the newly formed particles grew to 50 nm and **became** the*

**Reply:**

Thanks. It has been corrected.

**Reviewer:**

*Page 10 line 8*

*meaning that **more of** the  polymer-type organics **were** produced during the growth*

*of the new particles in this event.*

**Reply:**

Thanks. It has been corrected.

**Reviewer:**

*page 12 line 1 and following*

***Negative***  *biases, ranging from*

**Reply:**

Thanks. It has been corrected.

**Reviewer:**

*line 10*

*majority of the relative bias **is** still locate**d** between ....*

**Reply:**

Thanks. It has been corrected.

**Reviewer:**

*line 34*

 ***a consequence of*** *growth processes*

**Reply:**

Thanks. It has been corrected.

**Reference**

[revised manuscript text omitted]